# Application of HadCM3@Bristolv1.0 simulations of paleoclimate as forcing for an ice-sheet model, ANICE2.1: set-up and benchmark experiments

Constantijn J. Berends[1], Bas de Boer[1], Roderik S. W. van de Wal[1]

[1]Institute for Marine and Atmospheric research Utrecht, Utrecht University, The Netherlands

*Correspondence to*: Constantijn J. Berends (c.j.berends@uu.nl)

**Abstract.** Fully coupled ice-sheet-climate modelling over 10,000 – 100,000-year time scales on high spatial and temporal resolution remains beyond the capability of current computational systems. Forcing an ice-sheet model with pre-calculated output from a general circulation model (GCM) offers a middle ground, balancing the need to accurately capture both long-term processes, in particular circulation driven changes in precipitation, and processes requiring a high spatial resolution like ablation. Here, we present and evaluate a model set-up that forces the ANICE 3D thermodynamic ice-sheet-shelf model calculating the four large continental ice-sheets (Antarctica, Greenland, North America and Eurasia), with pre-calculated output from two steady-state simulations with the HadCM3 (GCM), using a so-called matrix method of coupling both components, where simulations with various levels of $pCO_2$ and ice-sheet configuration are combined to form a time-continuous transient climate forcing consistent with the modelled ice-sheets. We address the difficulties in downscaling low-resolution GCM output to the higher-resolution grid of an ice-sheet model, and account for differences between GCM and ice-sheet model surface topography ranging from interglacial to glacial conditions. Although the approach presented here can be applied to a matrix with any number of GCM snapshots, we limited our experiments to a matrix of only two snapshots. As a benchmark experiment to assess the validity of this model set-up, we perform a simulation of the entire last glacial cycle, from 120 kyr ago to present-day. The simulated eustatic sea-level drop at the Last Glacial maximum (LGM) for the combined Antarctic, Greenland, Eurasian and North-American ice-sheets amounts to 100 m, in line with many other studies. The simulated ice-sheets at LGM agree well with the ICE-5G reconstruction and the more recent DATED-1 reconstruction in terms of total volume and geographical location of the ice sheets. Moreover, modelled benthic oxygen isotope abundance and the relative contributions from global ice volume and deep-water temperature agree well with available data, as do surface temperature histories for the Greenland and Antarctic ice-sheets. This model strategy can be used to create time-continuous ice-sheet distribution and sea-level reconstructions for geological periods up to several millions of years in duration, capturing climate model driven variations in the mass balance of the ice sheet.

# 1 Introduction

Sea-level rise due to large-scale retreat of the Greenland and Antarctic ice-sheets poses one of the main long-term risks of climate change (Church et al., 2013). However, accurate projections of the magnitude and rate of retreat are limited by our understanding of the feedback processes between global climate and the cryosphere on centennial to multi-millennial time-scales. One way to test the performance of ice-sheet models that are used for these future projections, is to apply these models to ice-sheet evolution in the geological past, both during glacial periods with more ice than present-day, and warmer periods with less ice (e.g. Bamber et al., 2009; Pollard and DeConto, 2009; de Boer et al., 2013; Dutton et al., 2015).

Ideally, such a model set-up would consist of a general circulation model (GCM) fully coupled to an ice-sheet model, exchanging information every model time-step. However, whereas the computational load of typical ice-sheet models allows simulations of 10,000 – 100,000 years to be carried out within a reasonable amount of time, GCMs are much more computationally demanding, limiting simulation time to decadal or centennial time-scales. Fully coupled ice-sheet-climate modelling of complete glacial cycles is therefore not feasible with the current state of model infrastructure.

In order to gain insight into the long-term interactions between the climate and the cryosphere despite these computational limitations, different solutions have been proposed in the past. Several studies of past glacial cycles using ice-sheet models (Bintanja et al., 2002; de Boer et al., 2014) apply a present-day climate with a uniform temperature offset based on a "glacial index", usually from ice-core isotope records, adapting precipitation based on a Clausius-Clapeyron type relationship. Others have used a similar glacial index to create a linear combination of output of different GCM time-slice simulations (Marshall et al., 2000, 2002; Charbit et al., 2002, 2007; Tarasov and Peltier, 2004; Zweck and Huybrechts, 2005; Niu et al., 2017). Both types of studies share the shortcoming of having no clear physical cause for the prescribed climatological variations, and no explicit feedback from the cryosphere back onto the prescribed climate. Stap et al. (2014; 2016) used a zonally averaged energy balance model coupled to a one-dimensional ice-sheet model to simulate the behaviour of global climate and the cryosphere over millions of years, trading regional details for the ability to simulate long-term feedback processes. Others used dynamically coupled ice-sheet models to Earth System Models of Intermediate Complexity (Charbit et al., 2005; Ganopolski et al., 2010). This approach comes closer to the ideal case of an ice-sheet model fully coupled to a GCM, but since EMICs typically have a coarse spatial resolution, processes influencing the surface mass balance variably over the different parts of the ice-sheet (e.g. precipitation, ablation) still need to be parametrised. Other studies have asynchronously coupled ice-sheet models to GCMs (Herrington and Poulsen, 2012), or used fully-coupled ice-sheet-GCM set-ups with low-resolution GCMs for shorter periods of model time (Gregory et al., 2012), all showing that non-linear and non-local processes, particularly atmospheric stationary waves, surface albedo and altitude feedback, can significantly affect the behaviour of ice-sheets under a changing climate. Although such studies explicitly describe many more physical processes and feedbacks, computational resources quickly become a limiting factor in the length and number of simulations than can be performed. Abe-Ouchi et al.

(2013) performed a very detailed decoupling of the effects on climate of changes in $pCO_2$, albedo, surface elevation and atmospheric circulation based on several GCM snapshots and used these to force an ice-sheet model in a manner similar to both the glacial index method and the method described in this manuscript, highlighting the importance of isostatic adjustment of the lithosphere in producing the 100-kyr glacial cycles. By using pre-calculated GCM output, this approach makes it possible

to run many different simulations and investigate the effects of different physical processes.

The "matrix method" of hybrid ice-sheet-climate modelling (Pollard, 2010; Pollard et al., 2013) is based on a collection of steady-state GCM simulations where different values for one or more parameters such as $pCO_2$, insolation or global ice coverage are used to construct a so-called "climate matrix". By varying these parameters continuously over time and

interpolating between these pre-calculated climate states, a time-continuous climate history can be constructed, which can be used to force an ice-sheet model. Pollard et al. (2013) used this method to simulate the evolution of the Antarctic ice-sheet during the early Oligocene for 6 million years, using a 40km resolution ice-sheet model forced with output from the GENESIS version 3 GCM. They concluded that the method had some drawbacks, including a crude albedo feedback, and inability to smoothly track orographic precipitation, but that it was adequate for studying the large-scale ice-sheet evolution in which they

were interested.

An important difference between the glacial index approach and the matrix method is the latter's more explicit description of the feedback of an expanding or retreating ice-sheet on local, regional and global climate. In a glacial index model, the temporal evolution of the prescribed climatology is determined by an external forcing record (typically $pCO_2$, benthic $\delta^{18}O$ or ice-core

isotopes). The matrix method combines this external forcing with one or more internally modelled parameters (typically ice volume or extent) to determine the applied climatology, thus allowing changes in ice-sheet configuration to feed back on climate. Although this approach does still not explicitly describe all the feedback processes that can be included in a fully coupled ice-sheet model – AOGCM set-up, such as the influence of a growing ice-sheet dome on atmospheric circulation and stationary waves and the influence of freshwater fluxes on ocean circulation, it at least partially captures the feedbacks which

are not accounted for in a glacial index model and it does not not require much more computational resources.

In this study, we constructed a model set-up with a climate matrix consisting of two simulations with the HadCM3 GCM. The climate that is obtained from this matrix, based on the prescribed atmospheric $CO_2$ concentration and internally modelled ice-sheets, is applied to the mass balance module of the ANICE ice-sheet model, which simulates the evolution of all four major

continental ice-sheets (North America, Eurasia, Greenland and Antarctica) simultaneously. Difficulties in bridging the differences in model resolution and differences in ice-sheet configuration between GCM and ice-sheet model state, especially regarding the orographic forcing of precipitation resulting from ice-sheet advance, are addressed and overcome. As a benchmark experiment, a simulation of the entire last glacial cycle, from 120 kyr to present-day, was performed with this model set-up. We show that, because of several improvements to the way changes in albedo and precipitation are handled by

the model, we simulate ice-sheets at the Last Glacial Maximum (LGM) that agree very well with geomorphology-based reconstructions for Eurasia and better than previous ANICE versions for North America.

Previous work with the ANICE ice-sheet model (de Boer et al., 2013, 2014) used an inverse coupling method, where a global temperature offset is calculated in every model time-step such that the resulting deep-water temperature, combined with simulated global ice volume, matches a prescribed $\delta^{18}O$ record. This approach essentially determines how global climate should have behaved in order to produce the observed $\delta^{18}O$ record - regardless of what, if anything, could have caused the resulting strong, rapid climatic variations. Instead of working back from the a posteriori result of benthic $\delta^{18}O$, the new approach presented here starts with the a priori forcings of insolation and $pCO_2$ and determines what global climate should have looked like based on the forcings and the modelled ice-sheets. Although this still does not solve the discrepancy between the rapid cooling and sea-level drop suggested by the $\delta^{18}O$ record and sea-level data on the one hand, and the much more gradual decline in $pCO_2$ and surface temperature shown in the ice cores on the other that was observed by other studies (Bintanja and van de Wal, 2008; van de Wal et al., 2011; de Boer et al, 2014; Niu et al., 2017) it might provide new insights on the cause of this discrepancy.

## 2 Methodology

### 2.1 Climate model

HadCM3 is a coupled atmosphere-ocean general circulation model (Gordon et al., 2000; Valdes et al., 2017). It has been shown to be capable of accurately reproducing the heat budget of the present-day climate (Gordon et al., 2000) and has been used for future climate projections in the IPCC AR4 (Solomon et al., 2007) as well as palaeoclimate reconstructions such as PMIP2 (Braconnot et al., 2007) and PlioMIP (Haywood and Valdes, 2003; Dolan et al., 2011, 2015; Haywood et al., 2013). The atmosphere module of HadCM3 covers the entire globe with grid cells of 2.5 ° latitude by 3.75 ° longitude, giving a north-south resolution of about 278 km, whereas east-west resolution varies from about 70 km over northern Greenland (80 ° latitude) to about 290 km over southern Canada (45 ° latitude, the southern-most area covered by the ANICE grid). The ocean is modelled at a horizontal resolution of 1.25 ° by 1.25 °, with 20 vertical layers.

In their 2010 study, Singarayer and Valdes used HadCM3 to simulate global climate during the LGM, the pre-industrial period (PI) and several time slices in between. Orbital parameters representative of the era are used according to Laskar et al. (2004), atmospheric CO2 concentration is prescribed according to the Vostok ice-core record (190 ppmv at LGM; Petit et al., 1999; Loulergue et al., 2008) and orographic forcing follows the ICE-5G ice distribution reconstruction by Peltier (2004), shown in Fig. 1. Temperature and precipitation fields resulting from these two experiments are shown in Fig. 2 and Fig. 3.

The modelled glacial-interglacial global mean temperature difference is 4.3 K, which is in good agreement with results from other model studies (Hewitt et al., 2001; Braconnot et al., 2007), as well as reconstructions from multiple proxies (Jansen et al., 2007; Annan and Hargreaves, 2013). Comparisons of the model results with ice core isotope temperature reconstructions from Greenland (GRIP; Masson-Delmotte et al., 2005) and Antarctica (EPICA dome C; Jouzel et al., 2007), as well as

borehole-derived surface temperature reconstructions (Dahl-Jensen et al., 1998) indicate that glacial-interglacial temperature changes at these high latitudes are slightly underestimated by the model, by up to 1.5 K over Antarctica and up to 4 K over Greenland.

## 2.2 Ice-sheet model

To simulate the ice evolution on Earth we use ANICE, a coupled 3-D ice-sheet-shelf model (Bintanja and Van de Wal, 2008;

de Boer et al., 2013, 2014, 2015). It combines the shallow ice approximation (SIA) for grounded ice with the shallow shelf approximation (SSA) for floating ice shelves to solve the mechanical equations and incorporates a thermodynamical module to calculate internal ice temperatures. In ANICE, the applied mass balance is calculated using the parameterization by Bintanja and van de Wal (2005, 2008), which uses present-day monthly precipitation values, where changes in precipitation follow from a Clausius-Clapeyron relation as a function of free atmospheric temperature. Time- and latitude-dependent insolation

values according to the reconstruction by Laskar et al. (2004) are used to prescribe incoming radiation at the top of the atmosphere. Ablation is calculated using the surface temperature-albedo-insolation parameterization by Bintanja et al. (2002). In the transition zone near the grounding line, SIA and SSA ice velocities are averaged using the approach by Winkelmann (2011), as explained by de Boer et al. (2013). Sub-shelf melt is calculated based on a combination of the temperature-based formulation by Martin et al. (2011) and the glacial-interglacial parameterization by Pollard & DeConto (2009), tuned by de

Boer et al. (2013) to produce realistic present-day Antarctic shelves and grounding lines. A more detailed explanation is provided by de Boer et al. (2013) and references therein. Ice calving is treated by simple threshold thickness of 200 m, where any shelf ice below this thickness is removed. ANICE calculates ice sheet evolution on four separate grids simultaneously, covering the areas of the large Pleistocene ice-sheets: North America, Eurasia, Greenland and Antarctica. The areas covered by the four model domains are shown in Fig. 4. Horizontal resolution is 20 km for Greenland and 40 km for the other three

regions. Splitting North America and Greenland into separate model domains means the Laurentide and Greenland ice-sheets can no longer merge in the north, which they might have done during the LGM. However, we assume this to be not important for the large-scale evolution discussed in this study.

In their 2013 study, de Boer et al. simulated global ice distribution and sea level variation over the last 1 million years, forcing

ANICE with the LR04 benthic $\delta^{18}O$ record using an inverse routine. Their simulated LGM ice-sheets are shown in Fig. 5. They showed that their results are in good agreement with existing independent literature in terms of sea-level contributions (Rohling et al., 2009; Thompson and Goldstein, 2006), sea-water heavy isotope enrichment (Duplessy et al., 2002; Lhomme and Clarke, 2005) and other modelling studies (Huybrechts, 2002; Bintanja et al., 2005; Bintanja and van de Wal, 2008; Pollard

and DeConto, 2009), although ice-sheet location and extent do not agree well with evidence from geomorphology (Ehlers and Gibbard, 2007; de Boer et al., 2013 and references therein). The latter is likely a result from the absence of feedback from the growth of large ice-sheets onto large-scale atmospheric circulation patterns in the model, e.g. failing to reproduce the decrease in precipitation over the Barents Sea – Kara Sea area caused by the appearance of the large Fennoscandian ice dome, resulting
in the appearance of an unrealistically large ice dome over the Barents Sea. The highly parameterized climate forcing and resulting computational efficiency of ANICE allow these transient simulations of multiple glacial cycles to be carried out within 10 – 100h on single-core systems, making ensemble simulations feasible.

## 2.3 Climate matrix forcing

A climate matrix, as defined by Pollard (2010), is a collection of output data from different steady-state GCM simulations that
differ from each other in one or more key parameters or boundary conditions, such as prescribed atmospheric $pCO_2$, orbital configuration or ice-sheet configuration. At every point in time during the simulation, the location of the model state within this matrix is extracted from the matrix by interpolating between its constituent pre-calculated climate states. The pair of climate states generated by Singarayer and Valdes (2010) using HadCM3 is based on otherwise identical input parameters that differ in two respects: $pCO_2$ and ice-sheet coverage. These climate states can be viewed as points a two-dimensional climate
matrix, with $pCO_2$ constituting one dimension and ice-sheet coverage constituting another. In order to calculate a climate state for intermediate $pCO_2$ and ice-sheet coverage values, simple weight functions yielding linear interpolation in this climate phase-space will yield the corresponding monthly temperature and precipitation fields.

The weighting factor $w_{CO2}$ is calculated as:

$$w_{CO2} = \frac{pCO_2 - pCO_{2,LGM}}{pCO_{2,PI} - pCO_{2,LGM}}, \tag{1}$$

with $pCO_{2,PI}$ = 280 ppmv and $pCO_{2,LGM}$ = 190 ppmv. Although the dependence of radiative forcing on $pCO_2$ is logarithmic rather than linear, preliminary experiments showed that changing this in the calculation of the weighting factor did not result in significant changes in modelled sea-level at LGM, considering the uncertainty from other model parameters.

To determine the position of the model state along the $pCO_2$ dimension of the climate matrix, we use the EPICA ice core record by Lüthi et al. (2008). However, the ice-sheet coverage dimension of the matrix, described by $w_{ice}$, is more complicated and cannot be adequately described by a single scalar weight function. Since a continental-sized ice-sheet affects both local and global temperature mainly because of the increase in albedo, we chose to represent this process in the model by making
the ice-sheet coverage dimension of the climate matrix a spatially variable field $w_{ice}(x,y)$, calculated by scaling between the local absorbed insolation at present-day and at LGM. In this way the albedo feedback is captured more realistically. The

absorbed insolation $I_{abs}$ is calculated by multiplying incoming insolation at the top of the atmosphere $Q_{TOA}$ (from Laskar et al., 2004) with the surface albedo $\alpha$, the latter being calculated internally by ANICE:

$$I_{abs}(x,y) = \left(1 - \alpha(x,y)\right) \cdot Q_{TOA}(x,y). \tag{2}$$

5     The weighting field is calculated by scaling between the PI and LGM reference fields:

$$w_{ins}(x,y) = \frac{I_{abs,mod}(x,y) - I_{abs,LGM}(x,y)}{I_{abs,PI}(x,y) - I_{abs,LGM}(x,y)}, \tag{3}$$

running from 0 at the LGM to 1 for the PI. To account for both local and regional effects, a Gaussian smoothing filter F with a radius of 200 km, and a total average value are added to the weighting field:

$$w_{ice}(x,y) = \frac{1}{7}w_{ins}(x,y) + \frac{3}{7}F\left(w_{ins}(x,y)\right) + \frac{3}{7}\overline{w_{ins}}, \tag{4}$$

with the weights of the respective unsmoothed, smoothed and average values determined experimentally, such that the precipitation on the ice-sheet flanks, resulting from applying the Roe precipitation model, has values similar to those on the flanks of the ice-sheets in the reference GCM snapshots. The value of 200 km for the smoothing radius is based on de Boer et al. (2014), who used a similar smoothing procedure in their precipitation model. Preliminary experiments showed that changing this value did not result in significant changes in modelled LGM sea-level, within the uncertainty arising from other model parameters. For all four ice-sheets, these spatially variable ice weighting fields are combined with the scalar $pCO_2$ weight $w_{CO2}$ to yield the final weighting parameter $w_{tot}$:

$$w_{tot} = \frac{w_{CO2} + w_{ice}}{2}, \tag{5}$$

which is used to linearly interpolate between the states in the climate matrix and calculate the reference temperature, precipitation and orography. Preliminary experiments showed that changing the distribution of contributions from $w_{CO2}$ and $w_{ice}$ did not result in significant changes in modelled LGM sea-level, within the uncertainty arising from other model parameters. Since the two variables generally show coherent temporal behaviour, the two weighting factors are usually close together, meaning $w_{tot}$ doesn't change much when altering the distribution. When too much weight is given to $w_{ice}$ (between 2 and 4 times more than $wCO_2$), eventually a threshold is reached where the drop in $pCO_2$ during the early phase of the glacial cycle doesn't result in a strong enough cooling to trigger the growth of ice, thus breaking down this similarity.

Precipitation is customarily interpolated logarithmically to accurately reflect relative changes and to prevent the occurrence of negative values:

$$T_{ref,GCM}(x,y) = w_{tot} \cdot T_{PI}(x,y) + (1 - w_{tot}) \cdot T_{LGM}(x,y), \tag{6}$$

$$P_{ref,GCM}(x,y) = e^{\left(w_{tot} \cdot log(P_{PI}(x,y)) + (1 - w_{tot}) \cdot log(P_{LGM}(x,y))\right)}, \tag{7}$$

$$h_{ref,GCM}(x,y) = w_{tot} \cdot h_{PI}(x,y) + (1 - w_{tot}) \cdot h_{LGM}(x,y). \tag{8}$$

Being linear combinations of output data from a relatively low-resolution GCM, these three data fields necessarily have a lower resolution than the ice-sheet model to which they will be applied. To correct for this, the temperature and precipitation are adapted based on the difference between the interpolated reference orography $h_{ref,GCM}$ and the actual model orography, using the approach by de Boer et al. (2013) described in Appendix A.

Since the relative changes in ice-sheet size for Greenland and Antarctica are much smaller than those for North America and Eurasia, the relative changes in absorbed insolation in those regions are proportionally smaller and should therefore have had less impact on local climate. For example, for North America the total absorbed insolation over the model grid at LGM is 32 % lower than at present-day, whereas for Antarctica this change is only 5 %. This is reflected in the model by giving more weight to the $pCO_2$ parameter:

$$GRL, ANT: w_{tot} = \frac{3 \cdot w_{CO2} + w_{ice}(x,y)}{4}. \tag{9}$$

Preliminary experiments showed that here too, the sensitivity of the modelled ice volume to this distribution is relatively low.

**2.4 Lapse rate**

One of the major simplifications in the ANICE mass balance model is the assumption that temperature decreases linearly with
altitude - the spatially and temporally constant lapse-rate of -8 K/km. As has been shown by de Boer et al. (2014), the methodology of combining this constant lapse-rate with a global temperature offset derived from external forcing produced realistic results in terms of global and regional ice volume when simulating Pleistocene glacial cycles. However, even though the reference orography field obtained from the climate matrix is already close to the model orography and the correction applied to the GCM reference temperature field is therefore much smaller, preliminary experiments showed that even making
these relatively small corrections using a constant lapse-rate resulted in distorted results.

The limitations of this constant lapse rate procedure can be seen over the western part of Canada, an area that is hypothesized to have remained ice-free for the larger part of the last glacial cycle until a few thousand years before LGM. Here, results from the LGM experiment with HadCM3 (Singarayer and Valdes, 2010) indicate mean annual surface temperatures of around 235 K, or -38 °C. When calculating this surface temperature following the approach by de Boer et al. (2014), starting with the

present-day surface temperature at bedrock and scaling with the constant lapse-rate of -8 K/km to the ice-sheet surface (with an ice thickness of up to 5000 m, as indicated by ICE-5G), the resulting value is about 220 K, or -53 °C, about 15 degrees colder than calculated with the GCM, as shown in Fig. 6. A problem occurs during the inception and the subsequent build-up towards LGM, when this area is still ice-free in the model. Using the GCM-generated temperature field as a reference and scaling this down to bedrock level will then result in surface temperatures that are actually warmer than present-day. This is

unlikely and results in overestimated melt rates near the ice margins.

A solution to this is to slightly adapt the constant lapse-rate approximation. Assuming the GCM-generated temperature field at LGM is still based upon the present-day temperature field plus a global offset and a (local) lapse-rate correction, similar to the old ANICE method, this local lapse-rate correction field is then calculated as:

$$\lambda_{LGM}(x,y) = -\frac{T_{LGM}(x,y) - (T_{PI}(x,y) + \Delta T_{LGM})}{h_{LGM}(x,y) - h_{PI}(x,y)}, \tag{10}$$

, and the downscaling from the GCM grid to the ice model grid, previously described by Eq. A1, now being calculated as:

$$T(x,y) = T_{ref}(x,y) + \lambda_{LGM}(x,y)\Big(h(x,y) - h_{ref}(x,y)\Big), \tag{11}$$

where the local lapse-rate at LGM, $\lambda_{LGM}$, is calculated by dividing the difference between the local GCM-calculated surface temperature, $T_{LGM}$, and the extrapolated temperature at local bedrock altitude, $T_{bed}(x,y,t) = T_{PI}(x,y,t) + \Delta T_{LGM}$, by the change in local orography, $h_{LGM}$, with respect to present-day (hPI). The temperature offset $\Delta T_{LGM}$ is the mean difference in GCM-calculated temperature between the LGM and PI fields over the ice-free area in the respective model region (either North America or Eurasia) at LGM. For North America, this results in a value of $\Delta T_{LGM} = -14.9$ K. This methodology ensures that

when the modelled ice-sheet is identical to the ICE-5G ice-sheet at LGM and the $CO_2$ concentration is at the LGM value (190 ppmv pCO2), the temperature field that is used to calculate the mass balance is still identical to the GCM-calculated temperature field. It also guarantees that, when pCO2 is at 190 ppmv but no ice is present in the model, mean annual surface temperatures are uniformly lower than present-day by $\Delta T_{LGM}$.

Of course, the latter scenario only occurs during non-physical steady-state experiments such as forcing ANICE with the LGM GCM climate but initializing with present-day ice cover. During transient experiments, the modelled ice-sheets generally resemble those "expected" by the mass balance model through the climate state on which it is based, that the applied lapse-rate correction is generally small. This variable lapse-rate solution is used in the surface mass balance models for North America and Eurasia, since those regions see the dramatic changes in orography that require this correction. For Greenland and Antarctica, where the changes in ice cover are relatively small even during glacial cycles, the constant lapse-rate is still applied with a value of 8 K/km based on earlier work with ANICE by Helsen et al. (2013) and de Boer et al. (2014).

## 2.5 Precipitation

Present-day observations from Greenland indicate that the effect a continental-sized ice-sheet has on local precipitation is mostly due to geometry; more precipitation falls on the flanks due to orographic forcing, and as a result the dome becomes a plateau desert (Roe and Lindzen, 2001; Roe, 2002). The different character of this process calls for a different representation in the model than the absorbed insolation-based temperature calculation. In order to calculate monthly precipitation values, for North America and Eurasia we use the "local ice-weighting" method described by Pollard (2010). For every element of the spatial grid, ice thickness relative to the ice thicknesses at that element for the different reference GCM states, limited by the total volume of the ice-sheet, is used to obtain the interpolation parameter for the ice dimension of the climate matrix. Although physically, precipitation is influenced by surface altitude, not ice thickness, the fact that the weight is calculated based on scaling the model state between two extremes means the end result is the same as long as the rate of change of ice thickness and surface altitude is the same. The discrepancy between the two is caused by isostatic adjustment. During the inception phase of the glacial cycle, the ice grows slowly enough that there is hardly any discernible time lag between ice thickness and surface altitude. During the deglaciation this is not true anymore, but since ice-sheet evolution during that phase is dominated by ablation rather than precipitation, a parameterization based on elevation instead of ice thickness yields similar results. The interpolation parameter for the "ice" dimension of the climate matrix $w_{ice}$ is expressed as:

$$w_{ice}(x,y) = \frac{Hi_{mod}(x,y) - Hi_{PI}(x,y)}{Hi_{LGM}(x,y) - Hi_{PI}(x,y)} \cdot \frac{V_{mod} - V_{PI}}{V_{LGM} - V_{PI}}, \tag{12}$$

where $Hi_{mod}$ is the modelled local ice thickness and $Hi_{PI}$ and $Hi_{LGM}$ are the local ice thickness values in the reference fields from the GCM states. $V_{mod}$, $V_{PI}$ and $V_{LGM}$ are the modelled and reference ice-sheet volumes. For Greenland and Antarctica, only the total ice volume limitation is applied and the interpolation weight is calculated as:

$$w_{ice}(x,y) = \frac{V_{mod} - V_{PI}}{V_{LGM} - V_{PI}}. \tag{13}$$

The first term in Eq. 12 describes the local ice weighting method by Pollard (2010), whereas the second term describes the total ice volume scaling. Combining these two terms ensures that precipitation prescribed to the model only decreases over areas where the model actually simulates ice, and that the drop in precipitation caused by the ice-plateau-desert effect scales appropriately with ice-sheet size. Since the thickness of a growing ice-sheet levels off much earlier than its horizontal extent,

an ice-sheet of only a quarter of its LGM extent can already have nearly the same maximum thickness. Scaling precipitation based on local thickness alone will therefore result in the ice plateau becoming too dry too early in the growth phase, limiting further growth. Preliminary experiments showed that including the total ice-sheet volume in the calculation of the weighting factor solved this problem, resulting in a growth rate more in line with expectations from sea-level records.

The reason that the local ice thickness term is absent in the calculation for Greenland and Antarctica shown in Eq. 13 is that

the ICE-5G LGM ice-sheets that were used to calculate the corresponding GCM states are, in many places, thinner at LGM than at present-day, even though the total volume of the ice-sheet is larger. This would mean that an increase of modelled ice thickness would lead to an increase in applied local precipitation, causing unrealistic ice growth. Therefore, in order to prevent such unrealistic scenarios, precipitation is scaled only by the total ice-sheet volume.

For Greenland and Antarctica, the reference GCM precipitation field $P_{GCM,ref}$, is downscaled from the GCM to the ice-sheet model resolution based on the difference in temperature between the model state $T_{mod}$ and the reference GCM state $T_{GCM}$, as shown in Eq. 14, according to a Clausius-Clapeyron relationship, similar to the approach by de Boer et al. (2014) described in Appendix A. This ensures that smaller scale topographical features present in the model but not in the lower resolution GCM have an influence on local precipitation through their effect on local surface temperature.

$$P_{mod}(x,y) = P_{GCM,ref}(x,y) \cdot 1.0266^{(T_{mod}(x,y) - T_{GCM}(x,y))} \tag{14}$$

Similarly, for North America and Eurasia, precipitation is adjusted using the Roe (2002) parameterization for wind-orography-based correction of precipitation as described in Eq. A3 - A6, but now by using the GCM-generated precipitation and orography as reference fields instead of their ERA-40 equivalents. This allows for a better representation of orographic forcing of

precipitation on the migrating ice flanks as these ice-sheets advance and retreat, an effect that cannot be captured by interpolating by different GCM snapshots alone.

## 3 Results

### 3.1 Last glacial cycle benchmark

As a benchmark experiment, the new model set-up was used to perform a simulation of the last glacial cycle. The climate

matrix for this experiment consists solely of the PI_Control and LGM experiments by Singarayer and Valdes (2010). Following

the approach by Bintanja et al. (2002), the model was tuned by adjusting the ablation parameter $c_3$ in Eq. A9 individually for all four ice-sheet regions, such that their modelled sea-level contribution at LGM matched the values postulated by ICE-5G (Peltier, 2004). The resulting $c_3$ values, which are hereafter kept fixed, are shown in Table 1. This 120 kyr simulation took about 12 hours to complete on a single-processor system, meaning it is feasible to use this model set-up to perform ensemble

simulations without demanding excessive amounts of computation time.

Shown in Fig. 7 are the results of this experiment in terms of the global mean sea-level contributions of the four separate ice-sheets over time, as well as the total global mean sea-level, together with the same values from a simulation of the same period of time with the default ANICE model forced with the LR04 benthic $\delta^{18}O$ record using an inverse routine. As can be seen, the

new model set-up obtains a close match to the postulated ICE-5G LGM ice volume for all ice-sheets except Greenland. The resulting ice-sheets at LGM are shown in Fig. 8. As can be seen, the north-west Canadian corridor is now blocked by ice, which was still open in the default ANICE simulation shown earlier in Fig. 5. Although the main dome of the ice-sheets is not as thick as in the ICE-5G reconstruction, it now lies more westward than in the simulation with the default ANICE model, forming a ridge running from mid-west Canada to the eastern shores of Hudson Bay, which is in better agreement with the

reconstruction. The southern margin lies too far to the north, varying from 400 km near the Atlantic coast to up to 950 km in the mid-west. The Antarctic ice-sheet now shows a much stronger increase in ice volume around LGM, matching the 16 m of eustatic sea-level contribution postulated by ICE-5G (Peltier, 2004). Most of the ice mass increase takes place in West Antarctica - as can be seen, both the Ross and Ronne shelves become fully grounded. The Greenland ice-sheet does show some minor growth over the glacial cycle, though not as much as postulated. It must be noted that several modelling studies of

Greenland using the ANICE model (de Boer et al., 2013, 2014) have had trouble in this regard, mostly because of the difficulty in simulating the ice-shelves that might have formed around the continent at the time but are not there now (Bradley et al., 2018).

The simulated Eurasian ice-sheet is now in better agreement with the consensus regarding the Fennoscandian dome, as well

as with the total ice volume or sea-level contribution. When simulated with the default ANICE version, the main dome of the Eurasian ice-sheet forms over the Barents Sea, extending eastward to about 70°E. The new model set-up results in a dome over Fennoscandia and a smaller dome over the Barents Sea. The present-day southern North Sea area, formerly Doggerland, remains ice-free, in agreement with paleo data (Hughes et al., 2016). Compared to the recent DATED-1 reconstruction of the Eurasian ice-sheet (Hughes et al., 2016) at LGM shown in Fig. 9, the modelled ice-sheet does not extend as far south over

northern Germany, Poland and Lithuania. The simulated Atlantic side of the ice margin agrees well with the reconstruction, reaching the edge of the continental shelf everywhere.

Peltier (2004) provides an ice volume of the Eurasian ice-sheet of about 17 m sea-level equivalent based on GPS observations of isostatic rebound, whereas Hughes et al. (2016) state a volume of 24 m based on geomorphological evidence of the extent

and a logarithmic linear regression between ice sheet area and volume. By slightly increasing the ablation tuning parameter, thus decreasing ablation and increasing ice volume, we were able to produce a Eurasian ice-sheet with a volume of 24 m sea-level equivalent that matches the DATED-1 horizontal extent very well, as shown in Fig. 10. However, we believe that a "chain" of simulations such as this (an ice-sheet reconstruction, forcing a GCM, forcing an ice-sheet model) should aim for

consistency first, meaning that the ice-sheet produced at the end of the chain should match the one that was used as forcing at the start of the chain. Although there is more recent, more extensive data available for volume and extent of the Eurasian ice-sheet, prescribing to the ice-sheet model a climate that was calculated based on the presence of a different ice-sheet would make it much more difficult to determine the cause of any model-data mismatch in the final results. We therefore did not use this new, probably more physically realistic Eurasian ice-sheet as our benchmark.

**3.2 Sensitivity to forcing and model parameters**

In order to estimate the uncertainty in modelled global mean sea-level following from the uncertainty in the EPICA $pCO_2$ record, we performed simulations with the forcing record adjusted to its respective upper and lower bounds, based on an LGM uncertainty of 10 ppmv (Lüthi et al., 2008). Additionally, we investigated the model sensitivity to the four ablation tuning parameters $c_3$ for the different ice-sheets mentioned earlier by performing simulations where these parameters had been either

increased or decreased by 10% relative to their benchmark value. We also assessed model sensitivity to the SSA and SIA flow enhancement factors, with the upper and lower limits determined by Ma et al. (2010) in order to test the sensitivity to the ice sheet dynamics. Results from these different sensitivity tests are shown in Fig. 11. The resulting uncertainty in simulated LGM ice volume amounts to about 6 m sea-level equivalent in either direction, about 6 % of the total signal, for both the $CO_2$ and ablation parameter experiments. Sensitivity to the flow enhancement factor ratio is lower at about 4 % of the total signal.

**3.3 Benthic oxygen isotope abundance**

Included in ANICE is a module that tracks the oxygen isotope abundances of the ocean ($\delta^{18}O_{sw}$), precipitation and the ice-sheets. In the default ANICE version, an inverse routine is used to calculate a global temperature offset using the difference between modelled and observed benthic oxygen isotope abundance, implying that modelled and observed are per definition in agreement. In our new model set-up, the isotopic content of the ice-sheets is still tracked, but now the global mean temperature

anomaly from the climate matrix is used to determine a deep-water temperature anomaly ($\Delta T_{dw}$), and hence a modelled value for benthic $\delta^{18}O$. This deep-water temperature anomaly is calculated from the modelled mean annual surface temperature anomaly over the ocean following the approach by de Boer et al. (2014), using a 4,000-year running average and a scaling factor of 0.25. As opposed to the approach by de Boer et al. (2014), where an inverse method was used to match modelled benthic $\delta^{18}O$ to an externally prescribed record, modelled $\delta^{18}O$ can now be independently compared to such a record in order

to test the performance of the matrix method.

We compared our modelled benthic oxygen isotope abundance and the relative contributions to this signal by sea-water heavy oxygen enrichment and deep-water temperature change to data by Shakun et al. (2015), who analysed 49 ODP drilling locations where both surface-dwelling planktonic and benthic foraminiferal oxygen isotope abundance data were available, thereby allowing them to make a data-based decoupling of the contributions from ice volume and deep-water temperature to the benthic oxygen isotope signal. This model-data comparison is shown in Fig. 12. As can be seen, the results from the LGM benchmark experiment are in good agreement with the data, similar to the default ANICE model. The drop in benthic $\delta^{18}O$ at LGM of about 1.7 ‰ is reproduced comparably well by both the inverse method-forced model by de Boer et al. (2014) and the new matrix method-forced model set-up. The contribution from the change in deep-water temperature is slightly smaller in the new model set-up, though still in good agreement with the calculated global mean offset of 2 to 3 K at LGM. The new model set-up fails to reproduce the strong drop in benthic $\delta^{18}O$ during the inception of the glacial cycle, "catching up" at only 75 kyr.

### 3.4 Ice core temperature reconstructions

Shown in Fig. 13 are the modelled mean annual surface temperature anomalies over the Antarctic and Greenland ice sheets for the simulation with the default ANICE version and for the LGC benchmark experiment, compared to the EPICA Dome C reconstruction by Jouzel et al. (2007), and a stack of the GISP2 reconstruction by Alley (2000) and the NGRIP reconstruction by Kindler et al. (2014). As can be seen, both model versions agree well with each other and reasonably well with the Greenland isotope-based reconstructions (Alley, 2000; Kindler et al., 2014) regarding Greenland surface temperature anomalies. The Greenland records have been smoothed with a 4 ky running mean to filter out Dansgaard-Oeschger events, which are not present in our model forcing or climate reference runs and are also not included as feedback mechanisms in our model physics. Regarding Antarctic surface temperature anomalies, the new model set-up agrees particularly well with the EPICA isotope-based reconstruction (Jouzel et al., 2007), showing almost no significant deviations except for the first 20 kyr of the inception, where the model fails to reproduce the observed rapid cooling.

### 4 Conclusions

We have presented and evaluated a hybrid ice-sheet-climate model set-up that combines results from pre-calculated GCM simulations to force an ice-sheet model. Using the matrix method of GCM-ISM coupling, the impacts upon global climate of changes in atmospheric $CO_2$ concentration and global ice distribution are treated separately to construct a time-continuous climate forcing.

As a benchmark experiment, we used this new model set-up to simulate the entire last glacial cycle. Computational efficiency is such that this simulation could be performed within roughly 12 hours on a consumer-grade system. When compared with the default ANICE version by de Boer et al. (2014), the new model set-up performed better in simulating the volumes of the continental ice-sheets and their geographical position, and comparably well at simulating global mean deep-water temperature

and isotopic content. The improved performance in terms of geographical position is likely a result of the improved dynamically driven changes in precipitation as solved by the GCM. Niu et al. (2017) showed that forcing the PISM ice-sheet model with output from several different GCM simulations of LGM from PMIP3, all of which were prescribed the same initial ice-sheets, resulted in a wide range of ice-sheet sizes at LGM (50 to 150 m SLE). This illustrates that, even though the ice-sheet prescribed to the GCM leaves a clear local "fingerprint" in the resulting climate, especially in the simulated temperature, this is by no means a guarantee that forcing an ice-sheet model with that climate will reproduce an ice-sheet that resembles the ice-sheet in the boundary conditions.

Modelled temperature anomalies over Greenland and Antarctica agree well with ice-core isotope-based reconstructions. When accounting for uncertainty in the applied forcing and model parameters, the simulated volume of the four major continental ice-sheets (excluding contributions from smaller ice caps, glaciers, thermal expansion and ocean area changes) at LGM amounted to $97 \pm 11$ m sea-level equivalent ($\pm 2\sigma$ from the ensemble of simulations from the sensitivity analysis).

During the first 20 kyr of the inception, the model fails to reproduce the rapid drop in temperature and increase in ice volume visible in both benthic oxygen isotope records and ice-core isotope-based temperature reconstructions, implying that $pCO_2$ forcing alone is not sufficient to explain these observations without including some additional non-linear feedback processes. This is in line with results from other studies; studies like van de Wal et al. (2011) and de Boer et al. (2014) were able to reproduce the rapid cooling by using a forcing, such as a benthic oxygen isotope stack, that already incorporated the rapid decrease during the initial phase of the glacial cycle, whereas Bintanja and van de Wal (2008) and Niu et al. (2017) were unable to reproduce the rapid ice growth with $pCO_2$ forcing alone.

The effects of a growing ice-sheet on local and regional temperature are accounted for in the model through the resulting changes in albedo, but non-linear and non-local effects remain difficult to capture. Abe-Ouchi et al. (2013) constructed a model set-up similar to the matrix method presented here, but with more dimensions and corresponding GCM snapshots added to the matrix to decouple the different processes affecting temperature more explicitly: $pCO_2$, albedo, altitude and atmospheric stationary waves. Although their modelled ice-sheets at LGM do not match geomorphological reconstructions as well as the results presented here, they do report a stronger increase in ice volume during the inception. Expanding our climate matrix along the lines of their approach to more accurately describe the interplay between ice and climate for smaller ice-sheets could therefore potentially solve some of the repeatedly observed discrepancy between sea-level records and benthic $\delta^{18}O$ records on the one hand, and $pCO_2$ and temperature records on the other hand.

Other processes not accounted for in the albedo-based parameteristion of our climate matrix include glacial-interglacial changes in sea ice cover and changes in land albedo caused by changing vegetation. Including these feedback processes in the model could improve model performance in terms of the quantitative relation between $pCO_2$ and ice volume.

## 5 Code and data availability

NetCDF files containing output data from the benchmark simulation (ice thickness, bedrock topography, mean annual temperature, annual precipitation, albedo and surface mass balance) are available as online supplementary material at doi: 10.5194/gmd-2018-145supplement, or directly at https://zenodo.org/record/1288386.

The source code of ANICE2.1, including the new matrix method, is available online at doi: 10.5194/gmd-2018-145code, or directly at https://zenodo.org/record/1299522. Note that the model code can be compiled but cannot be run without input data describing present-day climate and topography, initial ice thickness and topography and GCM output files constituting the climate matrix. For any questions regarding ANICE, please contact c.j.berends@uu.nl.

The output of the HadCM3 experiments which we used to construct the climate matrix can be obtained from Paul Valdes at
the University of Bristol (P.J.Valdes@bristol.ac.uk).

## 6 Acknowledgements

The Ministry of Education, Culture and Science (OCW), in the Netherlands, provided financial support for this study via the program of the Netherlands Earth System Science Centre (NESSC). B. de Boer is funded by NWO Earth and Life Sciences (ALW), project 863.15.019. This work was sponsored by NWO Exact and Natural Sciences for the use of supercomputer
facilities. Model runs were performed on the LISA Computer Cluster, we would like to acknowledge SurfSARA Computing and Networking Services for their support. Special thanks go to Paul Valdes for sharing the data from his HadCM3 simulations with us.

## 7 Appendix A: Mass balance

In the ANICE version used by de Boer et al. (2014), the entire mass balance module is forced by a global temperature offset,
calculated from a prescribed $\delta^{18}O$ value and modelled global ice volume using the inverse routine by de Boer et al. (2013). This temperature offset, combined with a constant lapse-rate orography correction to account for changing ice thickness, is used to calculate a new monthly surface temperature field in every model time-step:

$$T(x,y) = T_{ref}(x,y) + dT_{glob} + \lambda\left(h(x,y) - h_{ref}(x,y)\right). \tag{A1}$$

Thus, the applied temperature $T$ at horizontal location $x, y$ is calculated at every model time step from the ERA-40 reference temperature $T_{ref}$, the global temperature offset $dT_{glob}$ and the difference between the model orography $h$ and the reference orography $h_{ref}$, multiplied by the constant lapse-rate $\lambda$ of -8 K/km. For Greenland and Antarctica, the applied precipitation $P$ is

then calculated by correcting the monthly present-day reference value $P_{ref}$ based on the difference between applied and reference temperature (Jouzel and Merlivat, 1984; Huybrechts, 1992):

$$P(x,y) = P_{ref}(x,y) \cdot 1.0266^{\left(T(x,y)-T_{ref}(x,y)\right)}. \tag{A2}$$

When simulating entire glacial cycles, the changes in ice-sheet geometry over North America and Eurasia are of a much larger scale then those over Greenland and Antarctica. In order to recreate the hypothesized westward growth of those ice-sheets during glacial inception, caused by orographic forcing of precipitation as moist wind blows up the slope of the ice-sheet and releases its moisture content, the precipitation model by Roe and Lindzen (2001) and Roe (2002) is used to calculate monthly precipitation values over these regions:

$$P(x,y) = P_{ref}(x,y)\frac{P_{Roe}(x,y)}{P_{Roe_{ref}}(x,y)}, \tag{A3}$$

$$dP_{Roe}(x,y) = e_{sat}(x,y)\,max\left(0,(a+bw'_{vv})\right)f(w'_{vv})dw'_{vv}, \tag{A4}$$

$$e_{sat}(x,y) = e_0 \cdot e^{\left(\frac{c_1(T(x,y)-T_0)}{c_2+T(x,y)-T_0}\right)}, \tag{A5}$$

$$f(w'_{vv}) = \frac{1}{N}e^{\left(-\left(\frac{w'_{vv}-w_0}{\alpha}\right)^2\right)}, \tag{A6}$$

$$w_{vv}(x,y) = max\left(0, W_x(x,y)\frac{\partial h(x,y)}{\partial x} + W_y(x,y)\frac{\partial h(x,y)}{\partial y}\right). \tag{A7}$$

Here, $e_{sat}$ is the saturation vapor pressure at the surface, which is a good proxy for the moisture content of the overlying air column. It is described by the Clausius-Clapeyron in Eq. A5 using the monthly mean surface temperature T, where $e_0 = 6.112$ mbar, $c_1 = 17.67$ and $c_2 = 243.5$ K. The vertical wind velocity $w_{vv}$ is calculated from the 850 Hpa wind and the surface gradient 20  according to Eq. A7. The precipitation $P_{Roe}$ is related to vertical wind velocity $w_{vv}$ through a probability distribution $f(w'_{vv})dw'_{vv}$, which is the probability that $w_{vv}$ lies between $w'_{vv}$ and $w'_{vv} + dw'_{vv}$, according to Eq. A6, where N is a normalisation factor and $\alpha = 1.15$ cm s$^{-1}$ is the measure of variability (Roe, 2002) in the vertical wind velocity. The precipitation $P_{Roe}$ is given by Eq. A4, where the constants a = $2.5 \cdot 10^{-11}$ kg$^{-1}$ s$^2$ m and b = $5.9 \cdot 10^{-9}$ s$^3$ kg were obtained by tuning to observations of Greenland (Roe, 2002). Eq. A4 is solved analytically using error functions (Roe and Lindzen, 2001).

Both $w_{vv}$ and $e_{sat}$ are calculated for both the reference state, using the reference temperature and orography fields, and for the model state, using the values at that model time step. The relative difference between the two modelled precipitation fields resulting from Eq. A4 is applied as an anomaly to the reference precipitation field to yield the applied precipitation field as described by Eq. A3.

Figures A1 and A2 show the mean annual temperature and total annual precipitation fields at present-day and LGM respectively, resulting from applying these two methods to the initial ERA-40 temperature and precipitation fields, using the difference between the reference ERA-40 orography and the modelled orography at present-day and LGM.

The monthly surface mass balance is calculated from the applied surface temperature and precipitation fields and the prescribed incoming radiation at the top of the atmosphere following Laskar et al. (2004). Monthly values for accumulation, refreezing and ablation are calculated separately and added. First, the snow fraction of precipitation is calculated according to the parameterisation by Ohmura (1999):

$$f_{snow}(x,y) = \frac{1-0.796\cdot\tan^{-1}\left(\frac{T(x,y)-T_0}{3.5}\right)}{2}, \tag{A7}$$

where the spatially variable monthly snow fraction $f_{snow}$ is defined as a function of 2-m air temperature. Monthly accumulation is simple the product of this fraction and monthly precipitation:

$$Acc(x,y) = P(x,y) \cdot f_{snow}(x,y). \tag{A8}$$

Local monthly ablation $Abl$ is parameterised as a function of the 2-m air temperature $T_{ano}$, albedo $\alpha$ and incoming solar radiation at the top of the atmosphere $Q_{TOA}$, following the approach by Bintanja et al. (2002):

$$Abl(x,y) = c_1(T(x,y) - 273.15) + c_2\left(Q_{TOA}(x,y) \cdot (1 - \alpha(x,y))\right) - c_3, \tag{A9}$$

with $c_1 = 0.0788$ m y$^{-1}$ K$^{-1}$, $c_2 = 0.004$ m$^3$ J$^{-1}$ and $c_3$ a tuning parameter different for each individual ice-sheet (tuned values listed in Table 1).

The local monthly refreezing $Refr$ is calculated from the available liquid water content $L_w$ (the sum of liquid precipitation and
ablation) and the superimposed water content $L_{sup}$, following the approach by Huybrechts and de Wolde (1999) and Janssens and Huybrechts (2000):

$$L_w(x,y) = P(x,y) \cdot \big(1 - f_{snow}(x,y)\big) + Abl(x,y), \tag{A10}$$

$$L_{sup}(x,y) = 0.012 \cdot \max\big(0, T_0 - T(x,y)\big), \tag{A11}$$

$$Refr(x,y) = \min\Big(L_w(x,y), L_{sup}(x,y), P(x,y)\Big). \tag{A12}$$

The surface mass balance *SMB* that will be used by the ice-sheet model is calculated as the sum of the accumulation *Acc*, the refreezing *Refr* and the ablation *Abl*:

$$SMB(x,y) = Acc(x,y) + Refr(x,y) - Abl(x,y). \tag{A13}$$

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

**Figure 1: LGM ice thickness distributions from the ICE-5G reconstruction (Peltier, 2004) for A) the Northern hemisphere and B) Antarctica. Contour lines for the Northern Hemisphere show ice thickness, contour lines for Antarctica show surface elevation. Bedrock elevtion where not covered by ice shown by colors, present-day shorelines shown in blue.**

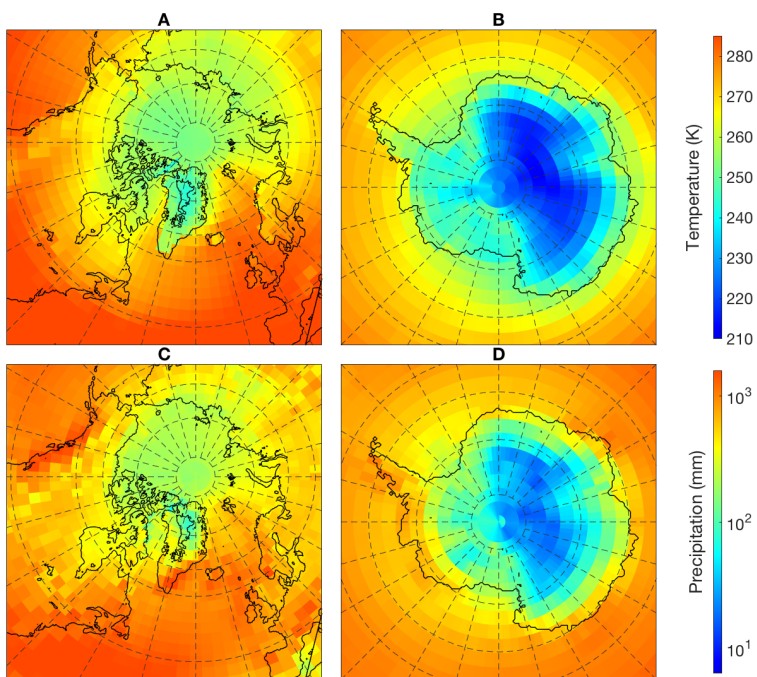

**Figure 1: Annual mean 2m temperature for the Northern Hemisphere (A) and Antarctica (B) and total annual precipitation (C and D), calculated with HadCM3 in the PI_Control experiment (Singarayer and Valdes, 2010).**

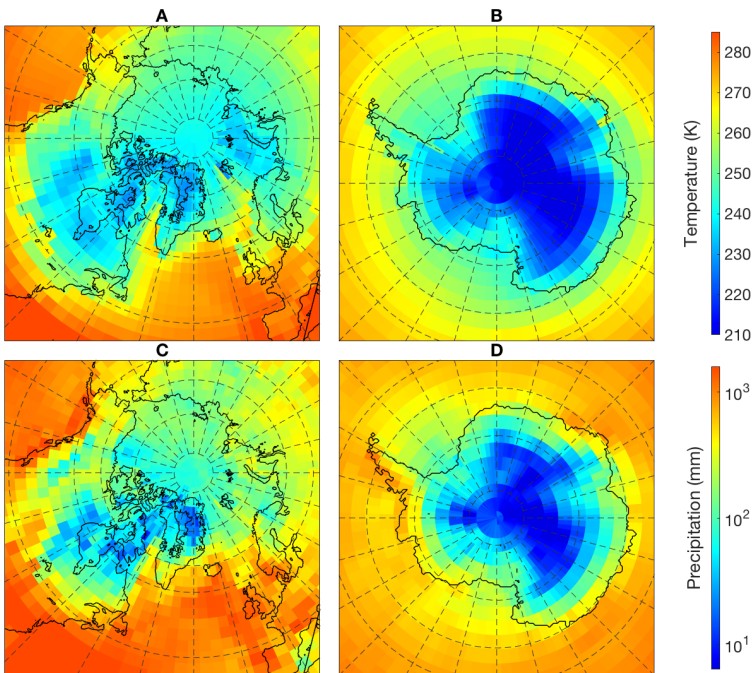

**Figure 3: Annual mean 2m temperature for the Northern Hemisphere (A) and Antarctica (B) and total annual precipitation (C and D), calculated with HadCM3 in the LGM experiment (Singarayer and Valdes, 2010).**

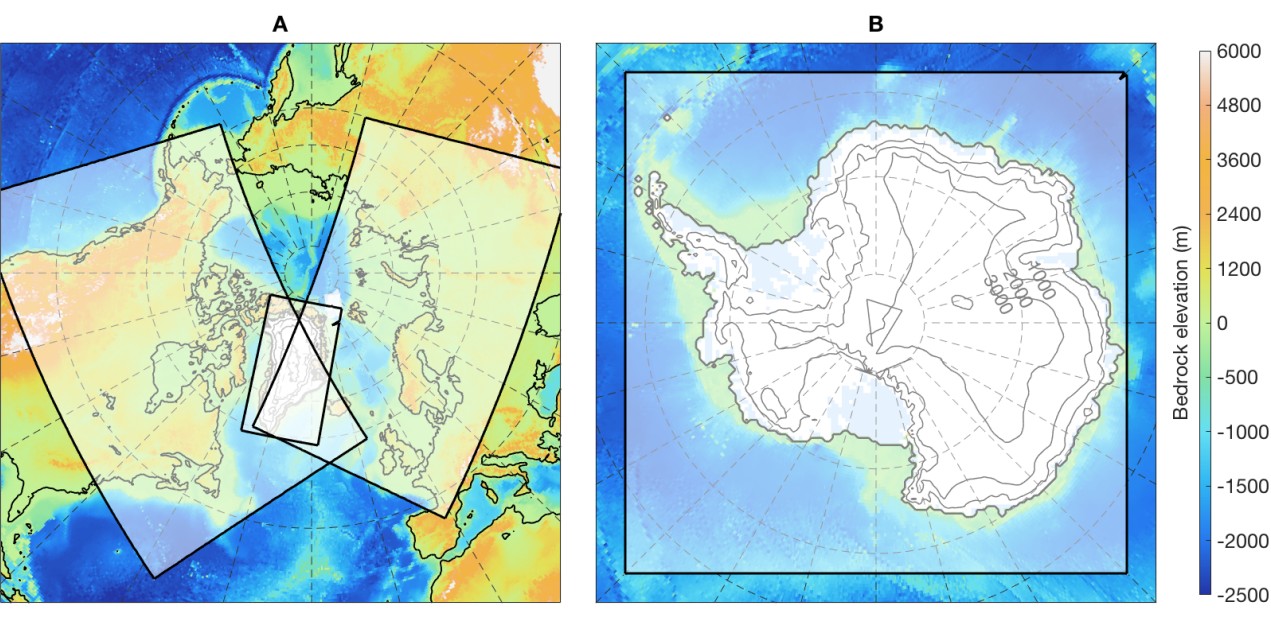

5    **Figure 4: The areas of the world covered by the four model domains of ANICE2.1. In the North America and Eurasia domains, Greenland is omitted.**

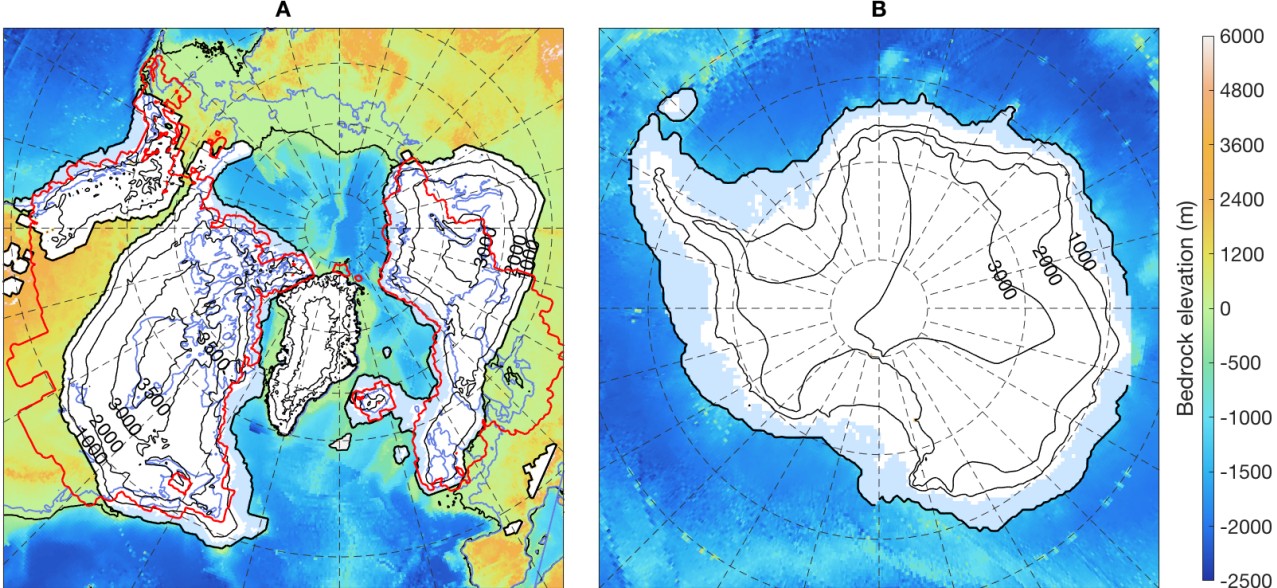

**Figure 5: Ice-sheets (white) and shelves (light blue) at LGM over A) the Northern Hemisphere and B) Antarctica, as simulated with the default ANICE version from de Boer et al. (2014). Contour lines for the Northern Hemisphere show ice thickness, contour lines for Antarctica show surface elevation. Bedrock elevtion where not covered by ice shown by colors, present-day shorelines shown in blue, ICE-5G ice margin shown in red.**

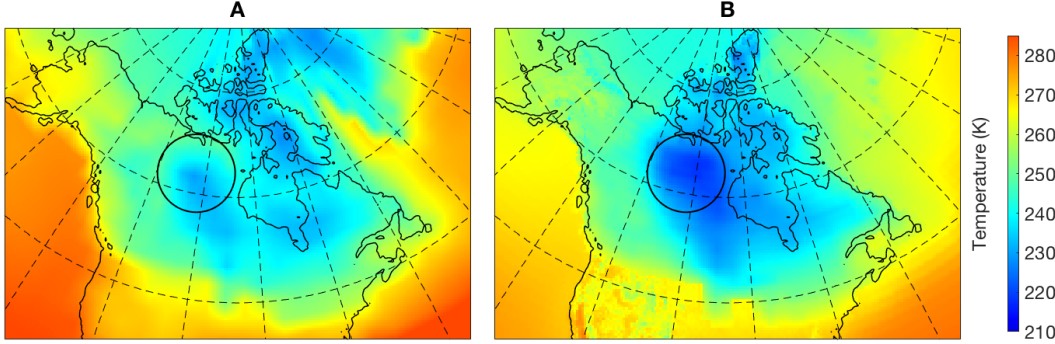

**Figure 6: Mean annual surface temperature at LGM over North America as generated with HadCM3 by Singarayer and Valdes (2010) (A) versus the temperature field generated for these conditions using a constant lapse-rate approach (B). GCM temperatures are substantially higher over the main dome of the ice-sheet (area indicated by black circle).**

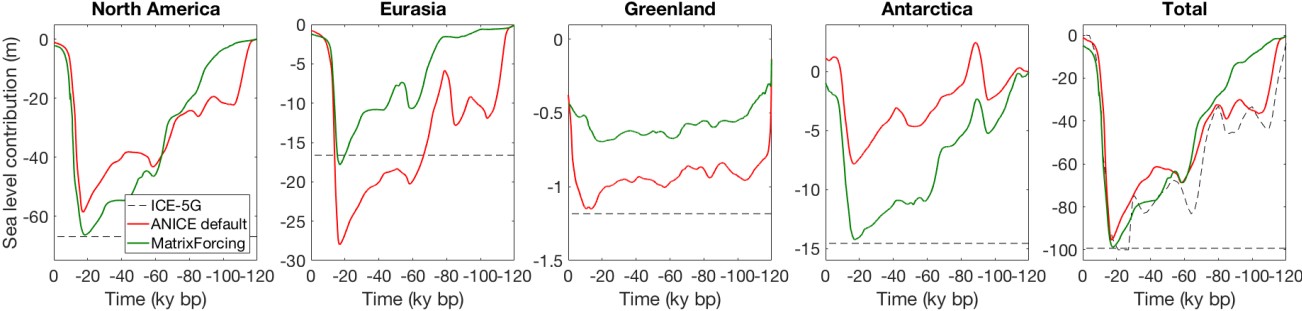

**Figure 7: Global mean sea-level contributions over time for the four individual ice-sheets, as well as the global total, for the LGC benchmark experiment (green) and the default ANICE control run (red), compared to the ICE-5G sea-level at LGM for the four individual ice-sheets, and throughout the last glacial cycle for the global sum (dashed line).**

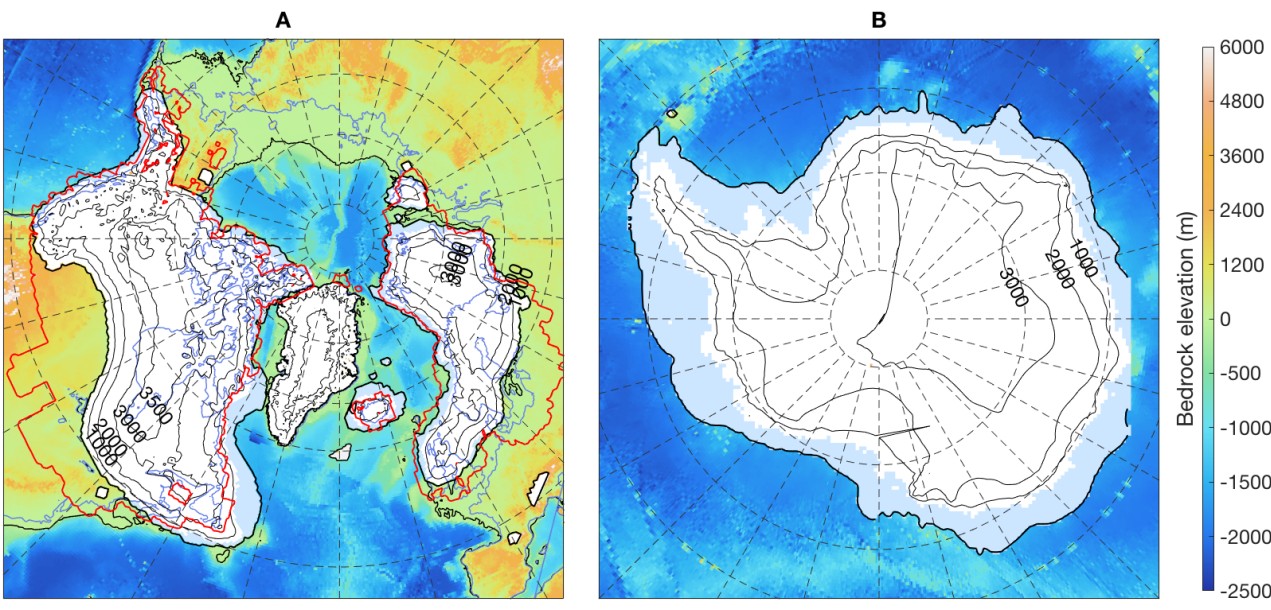

**Figure 8: Ice-sheets (white) and shelves (light blue) at LGM over A) the Northern Hemisphere and B) Antarctica, as simulated with the new model set-up. Contour lines for the Northern Hemisphere show ice thickness, contour lines for Antarctica show surface elevation. Bedrock elevtion where not covered by ice shown by colors, present-day shorelines shown in blue, ICE-5G ice margin shown in red.**

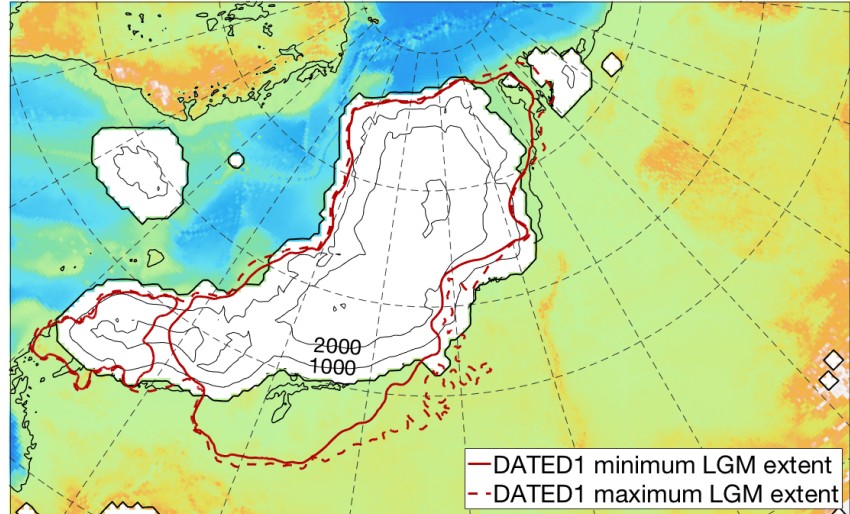

**Figure 9: Comparison of the simulated Eurasian ice-sheet at LGM with the DATED-1 reconstruction (Hughes et al., 2016). Contour lines show ice thickness. The modelled ice-sheet has a volume of 17 m sea-level equivalent, in agreement with the 17 m of the ICE-5G reconstruction, whereas the DATED-1 ice-sheet is equivalent to 24 m sea-level.**

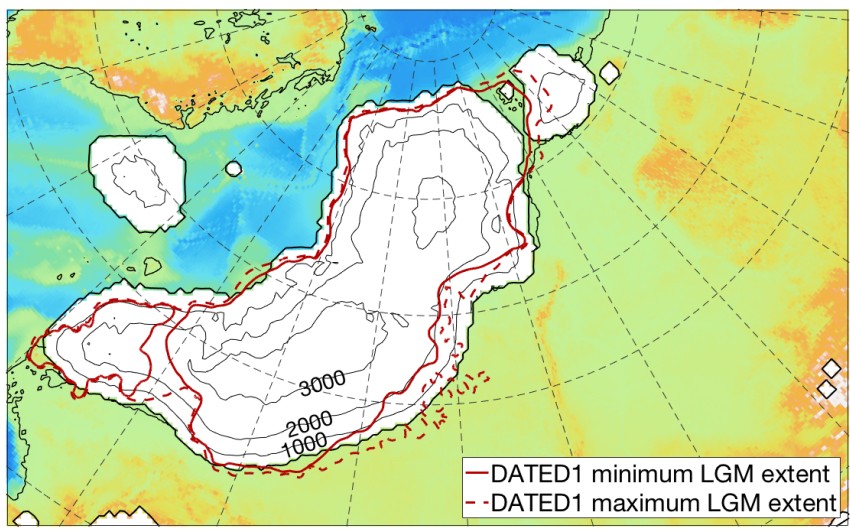

**Figure 10: Comparison of the larger simulated Eurasian ice-sheet at LGM with the DATED-1 reconstruction (Hughes et al., 2016). Contour lines show ice thickness. The modelled ice-sheet has a volume of 24 m sea-level equivalent, in agreement with the DATED-1 ice-sheet.**

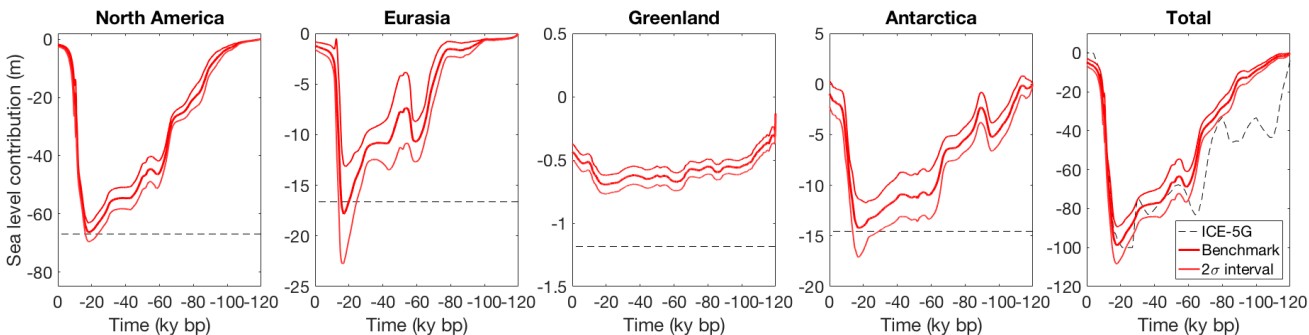

**Figure 11: Modelled sea level contribution over time for all four individual ice-sheets, and the total sum. The ± 2σ confidence interval is shown for the ensemble of simulations from the sensitivity analysis.**

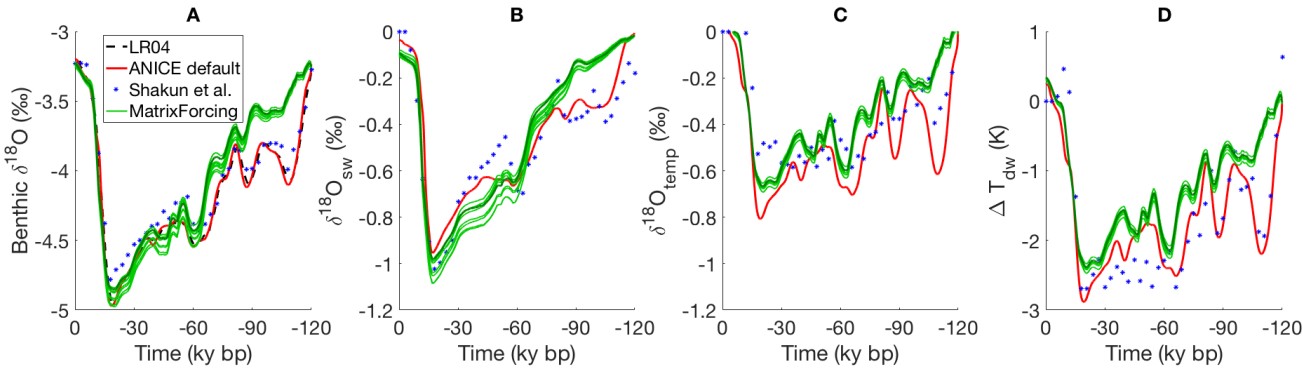

5  **Figure 12: A) modelled benthic oxygen isotope abundance from the default ANICE model (de Boer et al., 2014) and the LGM benchmark experiment compared to different datasets (LR04, Shakun et al. (2015). B) $\delta^{18}O$ of seawater due to depletion of heavy isotopes. C) contribution to benthic oxygen isotope abundance due to changes in deep-water temperature. D) derived deep-water temperature anomaly.**

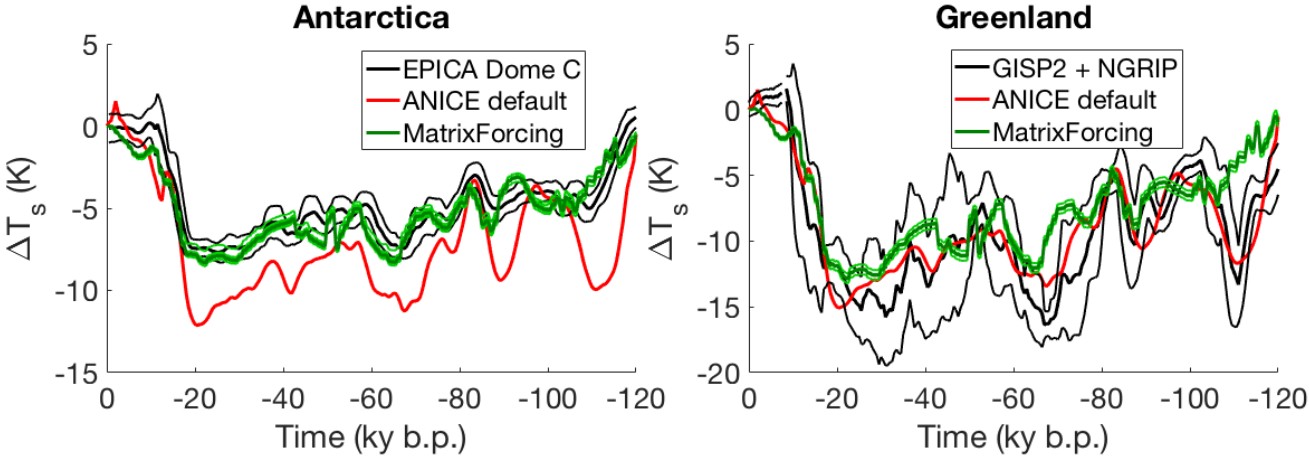

10  **Figure 13: Modelled versus reconstructed temperature anomaly for Antarctica (EPICA Dome C; Jouzel et al., 2007) and Greenland (GISP2; Alley, 2000; NGRIP; Kindler et al., 2014).**

**Table 1: Tuned values of the ablation parameter $c_3$ as used in Eq. A9.**

| **Region** | North America | Eurasia | Greenland | Antarctica |
|---|---|---|---|---|
| **$c_3$ (m/y)** | 0.14 | 0.23 | 0.19 | 0.14 |

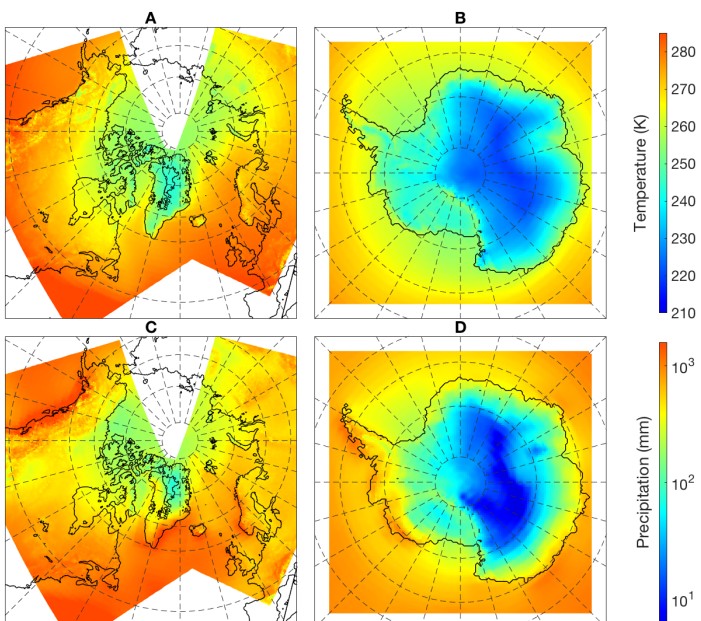

**Figure A1: Annual mean 2m temperature for the Northern Hemisphere (A) and Antarctica (B) and total annual precipitation (C and D), resulting from applying the constant lapse-rate temperature change and the Roe precipitation model to the ERA-40 climate fields.**

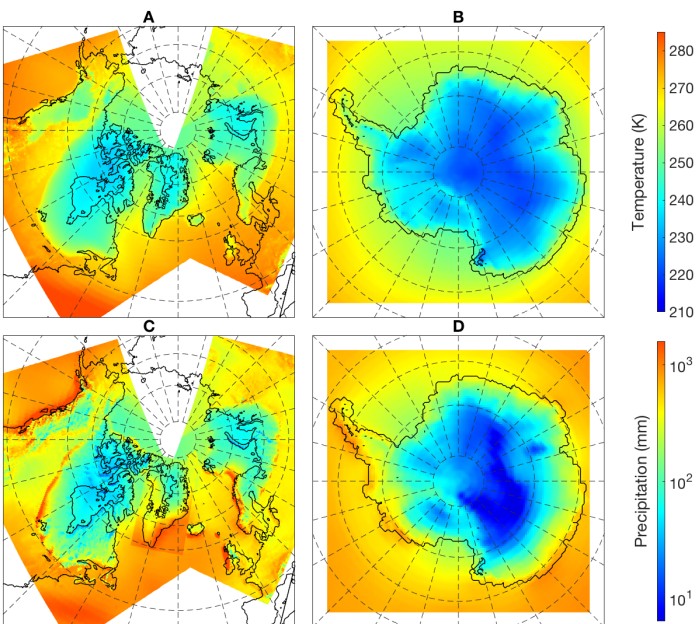

**Figure A2: Annual mean 2m temperature for the Northern Hemisphere (A) and Antarctica (B) and total annual precipitation (C and D), resulting from applying the constant lapse-rate temperature change plus global offset and the Roe precipitation model to the ERA-40 climate fields and the ANICE LGM ice-sheets.**