# Peer review of "Application of HadCM3@Bristolv1.0 simulations of paleoclimate as forcing for an ice-sheet model, ANICE2.1: set-up and benchmark experiments"

_Geoscientific Model Development, 2018_

## Referee Comment (RC1) · F. SAITO (Referee) · 31 Aug 2018

This paper describes the numerical ice-sheet model ANICE2.1, with coupling design of climate-model for 100kyr global ice-sheet simulation. Climate model part to interact with changes in ice-sheet distribution is computed using so-called matrix method, which is built from a series of pre-calculated outputs of a GCM (HadGCM).

I think this paper is fairly well written with some exception below, and can be accepted with minor revision.

Abe-Ouchi et al (2013) use a different approach to force an ice-sheet model, in which a

[Figure]

series of GCM snapshots are used to separate orbital, CO2, albedo etc effects on ice-sheet surface temperature. The method is not the same as two approaches (glacial index method and ESM coupling), and also not the same as the approach of the present paper. This study is limited to the northern hemisphere, but if the authors agree (I am not sure whether it is fair to tell this, because I am one of the authors of the paper), the authors may include the study as yet another example of hybrid GCM ice-sheet model application. In addition, several processes not included in the model are discussed in conclusion (around p13), which are discussed in Abe-Ouchi (2009, 2013).

As far as I understand, since ice sheet evolution is computed on the four separate regions, there is no chance to connect two ice sheets, e.g., Greenland and North America. In Fig. 4 the northwest part of Greenland seems to connect with NA ice sheet. I wonder how to handle this situation. Moreover, also in Fig. 4 or 7, simulated NA ice sheet extends on Eurasia. How to treat this? I suspect the model domain of NA ice sheet cover until East Siberia. Of course it is reasonable to assume that Siberia has been ice-free, in principle this is just an specification of the model of this paper. It is better to clarify these configuration. Possibly, it is enough to describe the four separated domain on the map.

Minor points.

p1 L10 'all ice' it too much. as far as I understand, neither glacier nor sea ice is included.

p3 L5 LGM should be defined here (now defined at L29).

p4 L4 degree C should be K.

p14 Eq A4, etc. write '\exp' instead of 'exp' if using LaTeX.

p15 L1. 2e-11, etc, should be written as $2 \times 10^{-11}$.

p15 L25 refer Table 1 after c3.

Fig.1. Need to describe the color as bedrock elevation where not covered by ice.

Fig.5,8,9. Need to describe the contour lines (thickness or surface elevation?)
SAITO Fuyuki.

---

## Referee Comment (RC2) · L. Tarasov (Referee) · 7 Sep 2018

**The inclusion of "GCM" in the title is mis-leading.**

with pre-calculated output from
several steady-state simulations with the HadCM3 general circulation model
**Inaccurate and misleading. Two simulations is not "several".**

The simulated ice-sheets at LGM agree well with the ICE-5G
reconstruction and the more recent DATED-1 reconstruction in terms of
total volume and geographical 20 location of the ice sheets.
**Since ICE-5G use DATED-1 precursors for margin constraint and**
**since the GCM was forced with ICE-5G boundary conditions, this**
**is a weak result**

Both types of studies share the shortcoming of having no clear
physical cause for the prescribed climatological variations,
**I would argue that the approach presented herein also has no clear**
**physical cause given the adhoc choice of weights and ignorance of all**
**the other feedbacks from ice sheets to climate..**

Others used dynamically coupled ice-sheet models to Earth System Models..
**Since you've started a list of alternatives, you should make it complete.**
**IE Should also consider asynchronous and accelerated coupling with GCMS, eg**
**Gregory et al, 2012, and Herrington and Poulsen, 2012.**

**on this same note, should also mention the option of using results from a**
**range of climate models, Eg Tarasov and Peltier, QSR 2004.**

Difficulties in bridging the differences in model resolution, as well
as other inconsistencies between model states, are addressed and
solved
**This is a vague arguable claim. Be more precise and accurate as to**
**what you do and do not "solve".**

the model, we simulate ice-sheets at LGM that agree very well with
geomorphology- based reconstructions
**This is not true for North America.**

This ensures the constructed climate history is in agreement with the
observed 15 pCO2 record and the modelled ice-sheet configuration,
thereby capturing the major feedback process between global climate
and the cryosphere, where any change in ice-sheet configuration has an
immediate impact on local climate through changes in albedo and
orographic forcing of precipitation
**This statement is not justified, especially with the use of only two**
**GCM climate snapshots. Atmospheric circulation and therefore climate**
**will depend non-locally on ice sheet geometry, a dependence that is not captured**
**by two or even a handful of GCM snapshots.**

It combines the shallow ice approximation (SIA) for grounded ice with the shallow
shelf
approximation (SSA) for floating ice shelves to solve the mechanical
**how are fluxes at the grounding-line handled?**

**How are sub-shelf melt and ice calving treated in this model?**

Horizontal resolution is 20 km for Greenland and 40 km for the other three regions
**For future work, I would recommend 20 km or finer grid resolution for non-**
ensemble best
**runs**

fig 4:
**please include present-day continental outlines even under ice using a different**
**colour than the black/grey contours for ice to aid geolocation**

strongly parameterized -> highly parameterized

should reference earlier work, eg EBM climate model coupling to ISMs

eq 1, linear co2 weighting factor
**given the near logarithmic depending of radiative forcing on pCO2,**
**justify why a linear dependence is imposed**

**eq 5: justify the equal weight contribution for Wco2 and Wice. Given**
**the large variation in insolation changes from the South to North of**
**eg the North American ice sheet complex over a glacial cycle, I**
**don't see how this constant weight mix makes sense.**

Gaussian smoothing filter F with
a radius of 200 km, and
**Why 200 km?**

Since the relative changes in ice-sheet size for Greenland and
Antarctica are much smaller than those for North America and Eurasia,
the changes in absorbed insolation in those regions should have less
impact on local climate. This is reflected in the model by giving more
weight to the pCO2 parameter
**So why not use this same weighting for the part of Canada covered**
**by the same latitudinal range as Greenland, especially given the**
**proximity of NorthWestern Laurentide/Innuitian ice sheets to Greenland?**
**Why not rely on the 200 km Gaussian radius to take care of the ice sheet**
**scale? I highly suspect that the need for this adhoc change is weighting**
**is due to the lack of accounting for larger scale (eg atmospheric dynamical)**
**effects of ice sheet on climate.**

eq 10
**Novel lapse rate approach that addresses a common problem especially**
**for those modellers who rely on a constant lapse rate value.**

eq 10
**Need to show equation for T(x,y,t) given T_{ref,GCM(x,y)} and lapse_LGM(x,y)**
**As I understand, eq A1 is for de Boer et al 2014, not this paper**
**(since a constant lapse rate is used)**

For Greenland and Antarctica, where the changes in ice cover are
relatively small even during glacial cycles, the constant lapse-rate
is still applied.
**justify 8K/km choice**

and that the drop in precipitation caused by the ice-plateau-desert
effect scales appropriately with ice-sheet size and that the drop in
precipitation caused by the ice-plateau-desert effect scales

appropriately with ice-sheet size
**what does "scales appropriately" mean? By what criteria?**

Similarly, for North America and Eurasia, precipitation is adjusted
using the Roe and Lindzen parameterization for wind orography- based
correction of precipitation as described in Eq. A3 - A6, but now by
using the GCM-generated precipitation and orography as reference
fields instead of their ERA-40 equivalents
**Why are no orography effects imposed on Greenland? Observed PD**
**fields show such effects**

Although the main dome of the ice-sheets is not
as thick as in the ICE-5G reconstruction
**this is a good thing**

Although the main dome of the ice-sheets is not
as thick as in the ICE-5G reconstruction, it now lies more westward than in the
simulation with the 5 default ANICE model,
which is in better agreement with the reconstruction
**Not clear where you main dome is given the 1000 m contour interval**

The Antarctic ice-sheet now shows a much stronger increase in ice
volume around LGM, matching the 16 m of eustatic sea-level
contribution postulated by ICE-5G (Peltier, 2004)
**Should reference more recent literature. The ICE-5G Antarctic**
**ice sheet has little constraint.**

However, since it was the ICE- 5G reconstruction that was used as
input for the HadCM3 simulation by Singarayer and Valdes (2004), we
aim to maintain 30 consistency and reproduce that particular ice-sheet
with our model rather than the DATED-1 LGM ice sheet.
**By what logic? You are assuming that ICE5-G is in conformity with**
**the GCM climate generated using ICE5-G boundary conditions. That is**
**a big assumption. The ice mask leaves a strong climate footprint and**
**so I would expect it not hard to match ICE5-G extent but matching I**
**see no rational to otherwise match ICE5-G topography**

pg 10 comparison to ICE5-G
**GCM fields generated with say ICE5-G boundary conditions will have a**
**strong imprint of the ice sheet margin on the resultant climate.  So**
**recreating ICE5-G ice extent with this interpolated climate forcing**
**offers little validation as to the utility of the approach. For me,**
**the challenge is to get a range of climates without the imprint of**
**assumed ice sheet boundary conditions used by the GCM.**

**fig 4 and 7**
**add the ICE5-G ice margin extent as say a red**
**contour to these plots to aid comparison**
**also use 500 m ice thickness contours to show more detail (1 km is awfully**
coarse)

The southern margin lies a little too far to the north
**This is an understatement. Be precise**

regarding Greenland surface temperature anomalies when neglecting the

strong negative excursions during Dansgaard-Oeschger events, which are
not present in our model forcing or 10 climate reference runs and are
also not included as feedback mechanisms in our model physics
**larger diffs than just missing D/O events in fg 12. Plot 4kyr**
**running mean and you'll see significant diffs.**

Fig 10
**Please replace this with a sensitivity parameter range that captures**
**say 90% confidence intervals for your parameters. Just switching**
**between PMIP III results from 2 different GCMs will from my**
**experience give a much larger spread in ice sheet volume**

Modelled temperature anomalies over Greenland and
Antarctica agree well with ice-core isotope-based reconstructions. When
**not for NGRIP**

Local monthly ablation Abl is parameterised as a function of the 2-m
air temperature Tano, albedo a and incoming solar radiation at the top
of the atmosphere QTOA, following the approach by Bintanja et
al. (2002):
...
with $c_1 = 0.0788$, $c_2 = 0.004$ and $c_3$ a tuning parameter different for each
individual ice-sheet.
**equations are dimensionally inconsistent and  need dimensional coefficients.**

These climate states span a two-dimensional climate matrix, with
**This is not what most modellers would take as a climate matrix**

calculated temperature between the LGM and PI fields over the ice-free area in the
region at LGM.
**specify region**

When accounting for uncertainty in the applied forcing and model
parameters, the simulated volume of the four major continental
ice-sheets (excluding contributions from smaller ice caps, glaciers,
thermal expansion and ocean area changes) at LGM amounted to 97 ± 6 m
sea-level equivalent.
**This shows that uncertainties are not adequately addressed. The uncertainties**
**in this modelled system (ie compared to "reality") are going to be much larger**
than 6 m SLE.

**At least 3 of the references to equations in the text have the wrong**
**equation number.**

**### review comments by PhD Candidate Taimaz Bahadory (in Lev Tarasov's group) to**
also address ####

P2-L14
  Still I'm not convinced how "This ensures the constructed climate
  history is in agreement with the observed pCO2 record and the
  modelled ice-sheet configuration".  All the climate states other
  than PI and LGM are interpolations based on some weights, so why
  should they be in agreement with the actual climatic history?  For
  instance if the jet-stream pattern variation would be a function of

a threshold in ice altitude, how would that be captured by
   interpolation?

P4-L21
   What does "some external forcing" mean?

P4-L28
   What is the "existing independent literature"?

P8-Eq. 11
   Why don't you use local altitude instead of the ice-thickness?  The
   difference at LGM could reach 1 km and it is surface elevation that physically
matters.

P8-L12
   Did you do the same calculation for lapse-rate over Greenland and
   Antarctica to check how small the difference would be?

P8-L16
   "Whereas a continental-sized ice-sheet influences temperature mainly
   through albedo"; is this true?  What about changes in atmospheric
   circulation, runoff and therefore changes in ocean circulation, and
   the elevation itself, hence the lapse-rate effect?

Fig. 6
   The total ice volume evolution, specially during the inception
   phase, doesn't follow the records; eg the 110 ka max volume.

**refs to add:**

Terminating the Last Interglacial: The Role of Ice Sheet–Climate Feedbacks
in a GCM Asynchronously Coupled to an Ice Sheet Model
ADAM R. HERRINGTON AND CHRISTOPHER J. POULSEN
DOI: 10.1175/JCLI-D-11-00218.1
2012

Modelling large-scale ice-sheet–climate interactions
following glacial inception
J. M. Gregory1,2, O. J. H. Browne1, A. J. Payne3, J. K. Ridley2, and I. C. Rutt4
Clim. Past, 8, 1565–1580, 2012
www.clim-past.net/8/1565/2012/
doi:10.5194/cp-8-1565-2012

---

## Author Comment (AC1) · 27 Sep 2018

Author comment replying to the referee comments by F. Saito

We'd like to thank the reviewer for their comments on the manuscript and would hereby like to address the concerns they raised.

In italics the comments, below our rebuttal. Page and line numbers refer to the revised manuscript.

*Abe-Ouchi et al (2013) use a different approach to force an ice-sheet model, in which a series of GCM snapshots are used to separate orbital, CO2, albedo etc effects on ice-*

*sheet surface temperature. The method is not the same as two approaches (glacial in-dex method and ESM coupling), and also not the same as the approach of the present paper. This study is limited to the northern hemisphere, but if the authors agree (I am not sure whether it is fair to tell this, because I am one of the authors of the paper), the authors may include the study as yet another example of hybrid GCM ice-sheet model application. In addition, several processes not included in the model are discussed in conclusion (around p13), which are discussed in Abe-Ouchi (2009, 2013).*

We agree that this is a very interesting and relevant study and will include a reference to it in the Introduction and Conclusions sections of our manuscript. P2, L27: Added a reference to the work by Abe-Ouchi et al. (2013). P14, L27: Added a few lines discussing the results reported by Abe-Ouchi et al. (2013).

*As far as I understand, since ice sheet evolution is computed on the four separate re-gions, there is no chance to connect two ice sheets, e.g., Greenland and North Amer-ica. In Fig. 4 the northwest part of Greenland seems to connect with NA ice sheet. I wonder how to handle this situation.*

It is true that the Greenland and Laurentide ice-sheets cannot connect in our model – in the North America module, the bedrock of Greenland has been manually lowered to well below sea-level, and vice versa in the Greenland module. This was done first by de Boer et al. (2013) to enable them to run the Greenland module at a higher resolution and more importantly with a mass balance module dedicated specifically for Greenland and one for the North America module. The large-scale behaviour of the two ice-sheets in terms of sea-level contribution, which is the main focus of our model, is not significantly affected by this and so we decided not to change this in our model version.

*Moreover, also in Fig. 4 or 7, simulated NA ice sheet extends on Eurasia. How to treat this? I suspect the model domain of NA ice sheet cover until East Siberia. Of course it is reasonable to assume that Siberia has been ice-free, in principle this is just*

*an specification of the model of this paper. It is better to clarify these configuration. Possibly, it is enough to describe the four separated domain on the map.*

Although this is not mentioned in the text, Figure 5 is the North America grid of the model. It does indeed cover the entire Bering Strait, as well as a very small portion of north-east Siberia. Since that area of the world is very dry, none of our simulations have ever encountered ice at the edge of the model grid. We will refer to a figure of the grid as presented in Fig. 3 in de Boer et al. (2014) which displays the same grid as used here. P5, L10: Added a reference to the relevant figure from de Boer et al. (2014).

*p1 L10 'all ice' it too much. as far as I understand, neither glacier nor sea ice is included.*

We agree that this statement is incorrect. We will correct this in the manuscript. P1, L10: changed the statement to correctly describe what ice is simulated by the model: "...thermodynamic ice-sheet-shelf model calculating the four large continental ice-sheets (Antarctica, Greenland, North America and Eurasia), ..."

*p3 L5 LGM should be defined here (now defined at L29).*

We agree, and will correct this in the manuscript. P3, L24: Added the definition of LGM P4, L16: Removed the definition of LGM

*p4 L4 degree C should be K.*

We agree, and will correct this in the manuscript. P4, L22: Changed degree C to K.

*p14 Eq A4, etc. write exp instead of exp if using LaTeX.*

We agree, and will correct this in the manuscript. P16, L17 (Eq. A5): Changed "exp(...)" to " e(...)".

*p15 L1. 2e-11, etc, should be written as 2 times 10ĚĘ-11.*

We agree, and will correct this in the manuscript. P16: Corrected all exponents.

*p15 L25 refer Table 1 after c3.*

We agree, and will correct this in the manuscript. P17, L25: Added a reference to Table 1 for values of parameter c3.

*Fig.1. Need to describe the color as bedrock elevation where not covered by ice. Fig.5,8,9. Need to describe the contour lines (thickness or surface elevation?)*

We agree, and will add this information to the figure captions. Fig. 1, 4, 5, 7, 8, 9: Added description of the colormap and contour lines to the figure captions.

---

## Author Comment (AC2) · 27 Sep 2018

Author comment replying to the referee comments by L. Tarasov and T. Bahadory

We'd like to thank the reviewers for their comments on the manuscript and would hereby like to address the concerns they raised.

The reviewers raised several questions about technical aspects of the model set-up presented in the manuscript, especially regarding the rationale behind the choices of different parameterisations. We agree that the manuscript could be improved at this

point and we clarified this in the new version, details are listed below. We'd like in particular to express our gratitude to the reviewers for pointing out an important detail – the fact that the stated dimensions of some of the parameters in the precipitation model were erroneous had indeed escaped our attention.

In Italics the comments, below our rebuttal. Page and line number refer to the new manuscript version.

*The inclusion of "GCM" in the title is misleading.*

We agree that the current title could be interpreted as meaning we constructed a coupled GCM-ISM. We will change the title to more accurately reflect our model set-up, where an ISM is forced with output from a GCM.

**P1, L1: changed the manuscript title.**

*with pre-calculated output from several steady-state simulations with the HadCM3 general circulation model Inaccurate and misleading. Two simulations is not "several".*

Although the method presented in this manuscript can be used to force the ice-sheet model with output from any number of GCM snapshots, the results presented here were indeed produced with only two snapshots. We agree that the statement is inaccurate, and will correct this.

**P1, L11: changed "several" to "two". P1, L16: clarified that the presented method can be applied to a matrix containing any number of GCM snapshots.**

*The simulated ice-sheets at LGM agree well with the ICE-5G reconstruction and the*

*more recent DATED-1 reconstruction in terms of total volume and geographical 20 lo-cation of the ice sheets. Since ICE-5G use DATED-1 precursors for margin constraint and since the GCM was forced with ICE-5G boundary conditions, this is a weak result*

*pg 10 comparison to ICE5-G GCM fields generated with say ICE5-G boundary con-ditions will have a strong imprint of the ice sheet margin on the resultant climate. So recreating ICE5-G ice extent with this interpolated climate forcing offers little validation as to the utility of the approach. For me, the challenge is to get a range of climates without the imprint of assumed ice sheet boundary conditions used by the GCM.*

As is shown in the referenced study by Niu et al. (2017), forcing a GCM with a certain prescribed ice-sheet and then using the resulting climate to force an ice-sheet model is by no means a guarantee that that ice-sheet model will produce the same ice-sheet that was initially prescribed to the GCM. Indeed, by doing exactly this, with different GCM's, Niu et al. (2017) produced ice-sheets at LGM ranging from 50 to 150 m SLE and in many cases exceeding the initially prescribed ice-sheet's extent by hundreds of kilometers. We will add a few lines to the manuscript discussing these results in order to clarify this point. While we agree with the reviewer that the "ultimate" model would require only orbital forcing in order to produce glacial cycles, accurately simulating all the feedbacks between ice-sheets and the atmosphere, the ocean, the carbon cycle and the biosphere, without being in any way "limited" by observations of the past, such a model is beyond the scope of this study.

**P14, L6: Added more context to the Conclusions section discussing the results presented by Niu et al. (2017).**

*Both types of studies share the shortcoming of having no clear physical cause for the prescribed climatological variations, I would argue that the approach presented herein also has no clear physical cause given the adhoc choice of weights and ignorance of*

*all the other feedbacks from ice sheets to climate..*

Although we do not claim that our model captures all existing feedback processes between the climate and the cryosphere, we do believe our model contains several important processes that are not represented in the "glacial index"-type models described in the statement. In these models, a temperature or climate forcing is prescribed based on an external forcing record, regardless of how the ice-sheets inside the model evolve. In our approach, we decouple the contributions to climate change caused by changes in pCO2 and changes in ice-sheet size plus insolation, calculating the latter based on the internal model state through the matrix method. We therefore believe our model set-up to be, though still not as comprehensive as a fully coupled GCM-ISM, at least more physically realistic than the glacial index model described in the statement, especially in the way spatial patterns of climate change are treated.

We will clarify the advantages of our approach in the manuscript.

We will address the reviewer's concerns regarding the "ad hoc choice of weights" later on, when he specifies exactly which choices of weights he finds to be insufficiently motivated.

**P3, L9: Added a short paragraph clarifying the difference between the glacial index approach and the matrix method.**

*Others used dynamically coupled ice-sheet models to Earth System Models.. Since you've started a list of alternatives, you should make it complete. IE Should also consider asynchronous and accelerated coupling with GCMS, eg Gregory et al, 2012, and Herrington and Poulsen, 2012. on this same note, should also mention the option of using results from a range of climate models, Eg Tarasov and Peltier, QSR 2004.*

We agree that these are valuable references. We will include them in the manuscript.

[Figure]

**P2, L17: Added a reference to Tarasov Peltier (2004) to the list of studies using a glacial-index method to force an ice-sheet model with output from different steady-state GCM simulations. P2, L25: Added a few lines discussing the work by Herrington Poulsen and Gregory et al. P18-22: Added these studies to the list of references.**

*Difficulties in bridging the differences in model resolution, as well as other inconsistencies between model states, are addressed and solved This is a vague arguable claim. Be more precise and accurate as to what you do and do not "solve".*

We agree that this statement should be more precise. We will change this in the manuscript.

**P3, L20: changed this line to more accurately describe which differences in GCM state and ice-sheet model state need to be accounted for.**

*the model, we simulate ice-sheets at LGM that agree very well with geomorphology-based reconstructions This is not true for North America.*

We agree that the phrase "very well" might be too optimistic for our simulation of North America. We will change this in the manuscript.

**P3, L25: differentiated between Eurasia and North America in our assessment of model performance.**

*This ensures the constructed climate history is in agreement with the observed 15 pCO2 record and the modelled ice-sheet configuration, thereby capturing the major*

[Figure]

*feedback process between global climate and the cryosphere, where any change in ice-sheet configuration has an immediate impact on local climate through changes in albedo and orographic forcing of precipitation This statement is not justified, especially with the use of only two GCM climate snapshots. Atmospheric circulation and therefore climate will depend non-locally on ice sheet geometry, a dependence that is not captured by two or even a handful of GCM snapshots.*

*P2-L14 Still I'm not convinced how "This ensures the constructed climate history is in agreement with the observed pCO2 record and the modelled ice-sheet configuration". All the climate states other than PI and LGM are interpolations based on some weights, so why should they be in agreement with the actual climatic history? For instance if the jet-stream pattern variation would be a function of a threshold in ice altitude, how would that be captured by interpolation?*

We did not intend to claim that the climate history constructed using our two-state climate matrix is a perfect representation of reality. The statement intends to illustrate how the changes in climate simulated by our model are all attributed directly to physical causes (changes in CO2, ice geometry and insolation) This is in contrast to the inverse forward models discussed earlier in the paragraph, which prescribe changes in temperature based on observations regardless of a clear physical cause. Neither do we claim that either method is better than the other; in the Conclusions section of the manuscript (p12-13) we discuss how the mismatch between data on sea-level and benthic oxygen isotopes on the one hand, both showing a rapid increase in ice volume during the early phase of the glacial cycle, and CO2 and temperature records on the other hand, both indicating a much less rapid cooling, is not solved by either type of model.

We agree that the statement might be interpreted as evidence of overconfidence in our methodology. We will correct this in the manuscript. We will also elaborate further on the various shortcomings and (over)simplifications of our method in the discussion section.

**P3, L31: Replaced the discussed statement with a new one that more accurately describes the differences between the inverse modelling approach and ours. P14, L27: Expanded the discussion of the various shortcomings of our climate parametrization (also in light of the concerns raised by the other reviewer)**

*It combines the shallow ice approximation (SIA) for grounded ice with the shallow shelf approximation (SSA) for floating ice shelves to solve the mechanical  how are fluxes at the grounding-line handled?  How are sub-shelf melt and ice calving treated in this model?*

In the transition zone near the grounding line, SIA and SSA ice velocities are combined using the approach by Winkelmann (2011), as explained by de Boer et al. (2013). Sub-shelf melt is calculated based on a combination of the temperature-based formulation by Martin et al. (2011) and the glacial-interglacial parameterization by Pollard  deConto (2009), tuned by de Boer et al. (2013) to produce realistic present-day Antarctic shelves and grounding lines. A more detailed explanation is provided by de Boer et al. (2013) and references therein. Ice calving is treated by simple threshold thickness of 200 m, where any shelf ice below this thickness is removed. We will add more information to the manuscript to clarify this.

**P5, L7: Added a few lines detailing the way grounding line fluxes, sub-shelf melt and calving are treated in the model.**

*Horizontal resolution is 20 km for Greenland and 40 km for the other three regions  For future work, I would recommend 20 km or finer grid resolution for non- ensemble best runs*

We thank the reviewer for this recommendation.

*fig 4: please include present-day continental outlines even under ice using a different colour than the black/grey contours for ice to aid geolocation*

We will add these outlines to the figure.

**Fig. 1, 4, 7: Added blue lines showing present-day shorelines to the relevant figures.**

*strongly parameterized -> highly parameterized*

We will correct this in the manuscript.

**P5, L25: Change "strongly" to "highly".**

*should reference earlier work, eg EBM climate model coupling to ISMs*

We will add more relevant references to the introduction section of the manuscript.

**P2, L19: Added references to work by Stap et al. with their EBM-ISM set-up.**

*eq 1, linear co2 weighting factor given the near logarithmic depending of radiative forcing on pCO2, justify why a linear dependence is imposed*

Several preliminary experiments, which we chose not to include in the manuscript, were dedicated to trying out various ways to translate changes in CO2, ice sheet geometry, surface albedo and other model variables into the weighting factors for the

climate matrix. We ran experiments with both a linear and a logarithmic dependence of the weighting factor wCO2 on pCO2 and concluded that the difference in outcome was negligible. We will add a line to the manuscript to describe these preliminary experiments and their influence on our choice of parameters.

**P6, L9: Added a line to describe these preliminary experiments.**

*eq 5: justify the equal weight contribution for Wco2 and Wice. Given the large variation in insolation changes from the South to North of eg the North American ice sheet complex over a glacial cycle, I don't see how this constant weight mix makes sense.*

We feel the reviewer might have misunderstood the way Wice is calculated in the model. This spatially variable weighting factor is calculated based on "absorbed insolation", the product of (1-albedo) and insolation. This links the changes in climate to the two components of this process: changes in insolation (an external forcing) and changes in albedo caused by advancing or retreating ice (a modelled variable) and thereby ensures that the large variations in absorbed insolation caused by the changes in the geometry of the ice-sheet complex are reflected in the calculations.

As with the previous comment, some preliminary experiments, which we chose not to include in the manuscript, were dedicated to finding proper values for the contributions of the two weighting factors. We clarified this in the text.

Sensitivity to the distribution was found to be relatively low; as can be seen from the results in the paper, the temporal evolution of ice volume and CO2 are very similar. This means the values of the two separate weighting factors are usually very close to each other, implying that assigning more weight to one or the other doesn't change the outcome much. Of course, when more weight is given to Wice, at some point the drop in CO2 during the inception doesn't decrease temperatures enough to trigger the inception any more. We will add a few lines to the manuscript to clarify this.
**P7, L9: Added more context to justify this choice of weights.**

*Gaussian smoothing filter F with a radius of 200 km, and Why 200 km?*

This value is based on earlier work with ANICE by de Boer et al., who used a similar smoothing algorithm to calculate changes in precipitation over the ice-sheet. As before, preliminary experiments not described in the manuscript investigated this parameter and found that results were not very sensitive to its value, so we chose not to change it. We will add a line to the manuscript to clarify this.

**P7, L1: Added a few lines to justify the choice of a 200 km smoothing radius.**

*Since the relative changes in ice-sheet size for Greenland and Antarctica are much smaller than those for North America and Eurasia, the changes in absorbed insolation in those regions should have less impact on local climate. This is reflected in the model by giving more weight to the pCO2 parameter So why not use this same weighting for the part of Canada covered by the same latitudinal range as Greenland, especially given the proximity of NorthWestern Laurentide/Innuitian ice sheets to Greenland? Why not rely on the 200 km Gaussian radius to take care of the ice sheet scale? I highly suspect that the need for this adhoc change is weighting is due to the lack of accounting for larger scale (eg atmospheric dynamical) effects of ice sheet on climate.*

The weighting factor Wice scales the absorbed insolation between two extremes: its maximum value at present-day and its minimum value at LGM. An LGM-sized ice-sheet will therefore always yield a weighting factor of 1 (meaning the GCM LGM simulation is used as forcing), regardless of the absolute change in absorbed insolation. For North America and Eurasia, where the continent changes from virtually ice-free to covered by vast ice-sheets, these changes are very large (a relative change of about 32

While we agree that a more elaborate approach, especially taking into account changes in North American ice sheet size into the calculations of Wice for Greenland and Eurasia, along the lines of Abe-Ouchi et al. (2013), would be more realistic, we chose to limit this first model set-up to only first-order effects.

We will add a few lines to the manuscript to clarify this.

**P7, L25: Added a few lines to justify the altered wice-wCO2 distribution for Greenland and Antarctica.**

*eq 10  Novel lapse rate approach that addresses a common problem especially  for those modellers who rely on a constant lapse rate value.*

We fully agree with the reviewer.

*eq 10  Need to show equation for T(x,y,t) given Tref,GCM(x,y) and lapseLGM(x,y)  As I understand, eq A1 is for de Boer et al 2014, not this paper  (since a constant lapse rate is used)*

We will add an equation to demonstrate how the new variable lapse-rate is used to calculate surface temperature.

**P9, L1: added this extra equation. P9, L3: fixed several typing errors.**

*For Greenland and Antarctica, where the changes in ice cover are relatively small even during glacial cycles, the constant lapse-rate is still applied.  justify 8K/km choice*

*P8-L12 Did you do the same calculation for lapse-rate over Greenland and Antarctica*

*to check how small the difference would be?*

We did not do the same calculation for Greenland and Antarctica. We only applied the variable lapse-rate to North America and Eurasia to solve an observed problem, i.e. mean annual surface temperatures increasing instead of decreasing towards LGM over ice-free areas, thus inhibiting ice growth. For Greenland and Antarctica such problems were never observed, so we never applied this method there. The choice of 8 K/km is based on earlier work with ANICE by de Boer et al. (2014) and with the regional climate model RACMO by Helsen et al. (2013). We will add a few lines to the manuscript to clarify this.

**P9, L18: added references to de Boer et al. (2014) and Helsen et al. (2013) to justify the 8 K/km choice.**

*and that the drop in precipitation caused by the ice-plateau-desert effect scales appropriately with ice-sheet size and that the drop in precipitation caused by the ice-plateau-desert effect scales appropriately with ice-sheet size what does "scales appropriately" mean? By what criteria?*

This statement only attempts to give a qualitative description. Preliminary experiments not described in the manuscript showed that if only the local ice thickness is used to calculate the weighting factor, precipitation decreases too fast over the main dome, because ice thickness reaches its peak value long before ice extent does. The resulting decrease in mass balance results in modelled ice-sheets that are far too small. By adding ice volume into the weighting factor calculation, precipitation decreases less quickly when the ice grows, allowing the ice-sheet to grow faster and reach its LGM size.

We agree that this is not clear in the manuscript right now. We will add a few lines to the manuscript to clarify this.

**P10, L9: Added a few lines explaining the rationale behind including total ice volume in the calculation of the weighting factor.**

*Similarly, for North America and Eurasia, precipitation is adjusted using the Roe and Lindzen parameterization for wind orography- based correction of precipitation as described in Eq. A3 - A6, but now by using the GCM-generated precipitation and orography as reference fields instead of their ERA-40 equivalents  Why are no orography effects imposed on Greenland? Observed PD  fields show such effects*

The Roe and Lindzen parameterization described in the manuscript is included in the model to account for changes in orography.  For North America and Eurasia this is important, because the flanks of the ice-sheet, where orographic forcing of precipitation occurs, move around over the continent as the ice sheets expand and retreat. For Greenland, the orographic changes are important for present-day but the changes throughout the glacial cycle are much smaller, as the ice flanks hardly migrates. The orographic forcing is already captured in the two GCM snapshots and it is therefore sufficient to use the interpolated states without requiring this correction.

We will add a few lines to the manuscript to clarify this.

**P10, L20-29: Clarified the explanation of the precipitation calculations and fixed Equation reference numbers.**

*Although the main dome of the ice-sheets is not as thick as in the ICE-5G reconstruction  this is a good thing*

We agree with the reviewer.

*Although the main dome of the ice-sheets is not as thick as in the ICE-5G reconstruction, it now lies more westward than in the simulation with the 5 default ANICE model, which is in better agreement with the reconstruction  Not clear where you main dome is given the 1000 m contour interval*

We agree that this is not clear from the figure. We will add a few extra contour lines to clarify this.

**Fig. 1, 4, 7: added an extra contour line at 3500m ice thickness to the relevant figures.**

*The Antarctic ice-sheet now shows a much stronger increase in ice volume around LGM, matching the 16 m of eustatic sea-level contribution postulated by ICE-5G (Peltier, 2004)  Should reference more recent literature.  The ICE-5G Antarctic  ice sheet has little constraint.*

*However, since it was the ICE- 5G reconstruction that was used as input for the HadCM3 simulation by Singarayer and Valdes (2004), we aim to maintain 30 consistency and reproduce that particular ice-sheet with our model rather than the DATED-1 LGM ice sheet.   By what logic? You are assuming that ICE5-G is in conformity with the GCM climate generated using ICE5-G boundary conditions. That is  a big assumption. The ice mask leaves a strong climate footprint and  so I would expect it not hard to match ICE5-G extent but matching I  see no rational to otherwise match ICE5-G topography*

While we agree with the reviewer that ICE-5G is hardly perfect and that there is more recent data available for both volume and extent, we believe that a chain of model simulations such as the one performed here (ice-sheet -> GCM -> climate -> ice-sheet

model -> ice-sheet) should aim for consistency first, i.e. the ice-sheet produced by the ice-sheet model should match the one that was prescribed to the GCM. Otherwise we'd be prescribing to the ice-sheet model a climate which was calculated based on a different ice-sheet, making it even harder to determine the cause of any observed model-data mismatches.

*fig 4 and 7 add the ICE5-G ice margin extent as say a red contour to these plots to aid comparison also use 500 m ice thickness contours to show more detail (1 km is awfully coarse)*

We agree that the requested elements would be of added value, and will add them to the figures.

**Fig. 4, 7: Added the ice-5g ice margin and a 3500 m ice thickness contour line.**

*The southern margin lies a little too far to the north This is an understatement. Be precise*

We agree that this statement is imprecise and overly optimistic. We will correct this in the manuscript.

**P11, L18: Clarified how far the modelled ice margin and ICE-5G margin lie apart.**

*Fig 10 Please replace this with a sensitivity parameter range that captures say 90 between PMIP III results from 2 different GCMs will from my experience give a much larger spread in ice sheet volume*

We will adapt the figure to make estimating the uncertainty in modelled sea-level arising from the uncertainty in our model parameters more intuitive.

**Fig. 10: Replaced lines of individual simulations with $\pm$ 2-sigma interval.**

*regarding Greenland surface temperature anomalies when neglecting the strong negative excursions during Dansgaard-Oeschger events, which are not present in our model forcing or 10 climate reference runs and are also not included as feedback mechanisms in our model physics  larger diffs than just missing D/O events in fg 12. Plot 4kyr  running mean and you'll see significant diffs.*

*Modelled temperature anomalies over Greenland and Antarctica agree well with ice-core isotope-based reconstructions. When  not for NGRIP*

We agree that the current way the icecore data was plotted made interpreting model-data differences difficult.  After subjecting the Greenland records to a 4 ky running mean, we find that modelled surface temperature anomalies fall within the high end of the +- 1 sigma range most of the time.

**Fig. 12: Merged the GISP2 and NGRIP records into a single stack. Subjected both the EPICA and Greenland stack records to a 4 ky running mean filter and added 4 ky window standard deviation range. P13, L18-21: changed the manuscript text to reflect the changes in the figure.**

*Local monthly ablation Abl is parameterised as a function of the 2-m air temperature Tano, albedo a and incoming solar radiation at the top of the atmosphere QTOA, following the approach by Bintanja et al. (2002): ... with $c_1 = 0.0788$, $c_2 = 0.004$ and $c_3$ a tuning parameter different for each individual ice-sheet.   equations are dimensionally*

*inconsistent and need dimensional coefficients.*

The reviewer is correct, there was a typo in the units of coefficient $\alpha$. We will correct this, and modify Equations A3 to A6 to be more in line with the original publication by Roe et al. (the current version in the manuscript describes the rather compliated analytical solution of a much more elegant integral).

**P15-17: Rewrote the equations described the Roe precipitation in their original, unintegrated form and expanded the explaining text.**

*These climate states span a two-dimensional climate matrix, with This is not what most modellers would take as a climate matrix*

While we agree that a climate matrix consisting of only two GCM snapshots is indeed rather small, we believe our method of model forcing has more in common with the matrix method than the glacial index method – especially because all the algorithms presented in the Methodology section are readily applicable to matrices consisting of more snapshots. However, we agree with the reviewer that stating that two points span a two-dimensional space is mathematically incorrect. We will correct this in the manuscript.

**P6, L1: Changed the relevant sentence to more accurately describe the climate matrix.**

*calculated temperature between the LGM and PI fields over the ice-free area in the region at LGM. specify region*

The region alluded to in this statement is the geographic area covered by the model grid. We will clarify this in the manuscript.

**P9, L5: Changed the relevant sentence to clarify which region is meant.**

*When accounting for uncertainty in the applied forcing and model parameters, the simulated volume of the four major continental ice-sheets (excluding contributions from smaller ice caps, glaciers, thermal expansion and ocean area changes) at LGM amounted to 97 $\pm$ 6 m sea-level equivalent. This shows that uncertainties are not adequately addressed. The uncertainties in this modelled system (ie compared to "reality") are going to be much larger than 6 m SLE.*

We did not intend to claim that the uncertainties in the applied $CO_2$ forcing and in our ice-sheet model parameters are the only sources of uncertainty in our sea-level reconstruction – merely that those are the only uncertainties that can be meaningfully investigated with this model set-up. We will clarify this difference in the manuscript.

**P14, L16: Clarified whence the uncertainty in the quoted number arises.**

*At least 3 of the references to equations in the text have the wrong equation number.*

We thank the reviewer for his attention to detail. We will correct the erroneous references.

**Entire manuscript: correct all erroneous equation and figure references.**

*P4-L21 What does "some external forcing" mean?*

The full statement, "Surface temperature is calculated from present-day monthly values, including a global temperature offset calculated based on some external forcing, and a constant lapse-rate orographic correction", refers to the way ANICE was used
by de Boer et al., Bintanja van de Wal. In these studies, ANICE was forced using the inverse coupling method, where a global temperature offset is calculated from a benthic oxygen isotope record, icecore isotope record or sea-level record. We agree that this is not clear. Since we do not use this method of forcing and do not allude to it any further, we will remove this statement from the manuscript.

**P5, L12: removed this statement.**

*P4-L28 What is the "existing independent literature"?*

de Boer et al. (2013) compared their results to other modelling studies (Huybrechts 2002, Pollard deConto 2009, Bintanja et al. 2005, Bintanja van de Wal 2008), geo-morphological evidence (Ehlers Gibbard 2007), sea-level records (Rohling et al 2009, Thompson Goldstein 2006) and the contribution of ice-sheets to sea-water heavy isotope enrichment (Duplessy et al 2002, Lhomme Clarke 2005). We will add more references to the manuscript to support our confidence in the ice-sheet model.

**P5, L18-22: included the references listed above in the manuscript.**

*P8-Eq. 11 Why don't you use local altitude instead of the ice-thickness? The difference at LGM could reach 1 km and it is surface elevation that physically matters.*

Since the ice thickness is scaled between two extremes (LGM value and zero) to calculate the weighting factor (which scales between 0 and 1), using surface elevation instead of ice thickness will yield the same result as long as the two variables change at the same rate. During the build-up phase of the glacial cycle this is generally true (the ice rarely grows faster than the lithosphere can adjust). During the deglaciation it is not, but since that process is dominated by ablation rather than accumulation, we

believe changing the parameterization from ice thickness to surface elevation will not significantly impact our results.

*P8-L16 "Whereas a continental-sized ice-sheet influences temperature mainly through albedo"; is this true? What about changes in atmospheric circulation, runoff and therefore changes in ocean circulation, and the elevation itself, hence the lapse-rate effect?*

We agree that this statement is incorrect – the changes in temperature are caused not only by changes in albedo but indeed also other processes, not all of which are captured by our model set-up. We will change this in the manuscript.

**P9, L20-23: Changed the statement so that it only illustrates why the choice was made to use a different parametrization for precipitation than the one for temperature without seeming to suggest that the current temperature parameterization captures all possible processes.**

*Fig. 6 The total ice volume evolution, specially during the inception phase, doesn't follow the records; eg the 110 ka max volume.*

The mismatch between our own modelled ice volume evolution and available records during the early part of the glacial cycle is discussed in the Conclusions section of the manuscript (page 12 – 13). We will add the ICE-5G pre-LGM eustatic sea-level record to Figure 6 to illustrate this mismatch.

**Fig. 6: Added the ICE-5G pre-LGM eustatic sea-level contribution record to the figure.**

---

## Author Response (AR2)

**Author's comment replying to editor's comments**

We'd like to thank the editor for reviewing our revised manuscript and would hereby like to address the concerns he raised. In italics the comments, below the changes we made to the manuscript in response to them.

*\* Title: "paleo" is a prefix, not a word. Please replace it with "paleoclimate" or similar.*

P4, L1: Changed "paleo simulations" to "simulations of paleoclimate" in the title.

*\* Abstract: Please revise the reference to "hybrid GCM-ice sheet modelling", as you acknowledge that your study does not fit into this category.*

P4, L8: Changed "hybrid GCM-ice sheet modelling" to "forcing an ice-sheet model with pre-calculated output from a GCM" in the abstract.

*\* Page 20, line 7: "climate models" is a very broad term, that could incorporate very fast models such as box models or EMICs. Please replace with "GCMs" or similar.*

P5, L8: Replaced "climate models" with "GCMs".

*\* Page 20, line 11: ... gain insight to the ...*

P5, L12: Corrected this mistake.

*\* Page 20, lines 24-30: Could you briefly summarise (~1-2 sentences) what these studies found/concluded? What were the strengths and weaknesses of these approaches?*

P5, L27 – P6, L2: Added a brief description of the main findings, advantages and limitations of these studies.

*\* Page 22, line 15: Insert "the" before "LGM" and "period" after "pre-industrial".*

P7, L23: Corrected these mistakes.

*\* Page 24, lines 8-9: "nearly identical" is a very vague statement. Please be more precise. Also I believe that, over this range, you can simply say "logarithmic rather than linear".*

P9, L21: Explained how changing the calculation of wCO2 from a linear to a logarithmic relation did not result in significant changes in modelled LGM sea-level.

*\* Page 24, line 26: Can you justify the choice of weights in Equation 4?*

P10, L10: Briefly explained how these weights were determined experimentally.

*\* Page 25, line 1: Again "quite low" is a vague value statement. Please be more precise.*

P10, L21: Amended this statement.

*\* Page 25, lines 7-11: Again, please be more precise throughout this section. Use a more specific statement than "quite low". What is the threshold above which the behaviour that you describe occurs?*

P10, L23: Clarified this threshold in the manuscript.

*\* Page 29, line 21: ... increase that takes place ...*

P15, L12: Corrected this mistake.

*\* Page 31, line 13: ... "catching up" at only ...*

P17, L5: Corrected this mistake.

*\* Figure 6: Please revise the caption to include the ICE-5G record that you have added to the final panel.*

Fig. 6: Revised the caption.

*RESPONSES TO REVIEWERS*

*\* Page 1, lines 23-27: This information would be highly beneficial to the reader. Please incorporate it into the manuscript itself.*

P8, L22-25: Added this information to the manuscript.

*\* Page 2, lines 1-5: Please incorporate this information into the manuscript (into the main text and/or figure captions) rather than requiring the reader to consult Boer et al. (2014).*

Figure 4: Added an additional figure depicting the four model domains and added a reference to this figure in the text.

*\* Page 5, lines 27-34: Your response indicates known limitations to your approach i.e. you do not claim that it captures all existing feedback processes, and you do not claim that it is as comprehensive as a fully-coupled GCM. This is fine, but please incorporate these caveats into the text that you have added at page 21, lines 8-13.*

P6, L19-22: Added a brief description of the caveats of our model approach with respect to a fully coupled ice-sheet model – AOGCM to the manuscript.

*\* Page 14, lines 11-15: Please incorporate this information into the manuscript, as it provides significant justification for your choice of methodology.*

P15, L32 – P 16, L4: Extended the justification for using the ICE-5G tuned result as a benchmark instead of the DATED-1 tuned result.

*\* Page 18, lines 4-9: Again, this is important information and should be incorporated into the manuscript.*

P13, L11-17: Added a justification for using ice thickness rather than surface elevation as a basis for calculating precipitation to the manuscript.

[revised manuscript text omitted]